# Time-resolved mapping of genetic interactions to model rewiring of signaling pathways

**Florian Heigwer**[1,2,3], **Christian Scheeder**[1,3,2], **Thilo Miersch**[1,3], **Barbara Schmitt**[1,3], **Claudia Blass**[1,3], **Mischan Vali Pour Jamnani**[1,3], **Michael Boutros**[1,3]*

[1]Division Signaling and Functional Genomics, German Cancer Research Center, Heidelberg, Germany; [2]HBIGS Graduate School, Heidelberg University, Heidelberg, Germany; [3]Department of Cell and Molecular Biology, Medical Faculty Mannheim, Heidelberg University, Heidelberg, Germany

**Abstract** Context-dependent changes in genetic interactions are an important feature of cellular pathways and their varying responses under different environmental conditions. However, methodological frameworks to investigate the plasticity of genetic interaction networks over time or in response to external stresses are largely lacking. To analyze the plasticity of genetic interactions, we performed a combinatorial RNAi screen in *Drosophila* cells at multiple time points and after pharmacological inhibition of Ras signaling activity. Using an image-based morphology assay to capture a broad range of phenotypes, we assessed the effect of 12768 pairwise RNAi perturbations in six different conditions. We found that genetic interactions form in different trajectories and developed an algorithm, termed MODIFI, to analyze how genetic interactions rewire over time. Using this framework, we identified more statistically significant interactions compared to end-point assays and further observed several examples of context-dependent crosstalk between signaling pathways such as an interaction between Ras and Rel which is dependent on MEK activity.

**Editorial note:** This article has been through an editorial process in which the authors decide how to respond to the issues raised during peer review. The Reviewing Editor's assessment is that all the issues have been addressed (see decision letter).
DOI: https://doi.org/10.7554/eLife.40174.001

*For correspondence:
m.boutros@dkfz.de

**Competing interests:** The authors declare that no competing interests exist.

## Introduction

Gene-gene interactions – the epistatic influences of one gene's effect on the function of another gene – have widespread effects on cellular and organismal phenotypes, ranging from fitness defects in unicellular organisms to interactions between germline and somatic variants in cancer (*Baryshnikova et al., 2013*; *Billmann and Boutros, 2017*; *Boone et al., 2007*; *Burgess, 2016*; *Carter et al., 2017*; *Ideker and Krogan, 2012*; *Mani et al., 2008*; *Phillips, 2008*; *Taylor and Ehrenreich, 2015*). In past studies, statistical genetic interactions (also simply referred to as genetic interactions) have been defined as an unexpected phenotypic outcome observed upon simultaneous perturbations (or knock-outs) of two genes that cannot be explained from the genes' individual effects (*Beltrao et al., 2010*; *Fisher, 1930*; *Mani et al., 2008*).

Genetic interactions can be discovered using pairwise perturbations of genes, a strategy which has been experimentally used at large scale in yeast (*Collins et al., 2007*; *Costanzo et al., 2010*; *Fiedler et al., 2009*; *Tong et al., 2001*), *C. elegans* (*Lehner et al., 2006*), *Drosophila* (*Fischer et al., 2015*; *Horn et al., 2011*), *E. coli* (*Babu et al., 2011*) and human cells (*Kampmann et al., 2013*; *Laufer et al., 2013*; *Roguev et al., 2013*; *Shen et al., 2017*). To create genetic interaction maps,

**eLife digest** Within a cell, communication routes that involve many different genes work to control how the cell responds to the environment. Although different communication routes – so called signaling pathways – control different cell activities, they do not work in isolation. Instead, they form part of larger regulatory networks that maintain the cell in an appropriate state. As such, changing the activity of one pathway may in turn affect another seemingly unrelated pathway.

The Ras signaling pathway helps to control when cells divide. When this signaling is not regulated correctly, cells can start to divide uncontrollably, leading to cancer. Drugs that suppress the activity of overactive Ras pathways could help to treat cancer. But how do the wider regulatory networks in the cell rewire themselves over time in response to this treatment?

To investigate this question, Heigwer et al. used a method called RNA interference to alter the activity of different pairs of 168 genes in fruit fly cells that had been grown in the laboratory. This meant 12,768 gene interactions were examined in total. Some of the cells had been treated with a drug that suppresses Ras signaling. By developing a new cell imaging and analysis system, Heigwer et al. could examine how the cell's regulatory networks were affected by the drug at three different time points after treatment. The results show that housekeeping genes, which handle basic cell duties, take more time to rewire their interactions than signaling pathways.

Heigwer et al. also developed a computational method – called MODIFI – to analyze how environment and time affect how genes interact. This highlighted a number of signaling pathways that are strongly affected by the suppression of Ras signaling, including an unexpected immune signaling pathway.

In the future, more research will be needed to study the context-dependency of interactions between genetic networks in different cell types and in living organisms. A better understanding of this context-dependency will be important to understand how cancerous cells develop drug resistance. The data collected by Heigwer et al. could also be used by other researchers to explain any unexpected gene interactions that affect the signaling pathways they are studying.
DOI: https://doi.org/10.7554/eLife.40174.002

these studies systematically identified alleviating (e.g. better fitness than expected) or aggravating (e.g. worse fitness than expected) genetic interactions, which can then be used to generate 'genetic interaction profiles' for each gene. Several studies have shown that genes involved in the same cellular processes have highly similar genetic interaction profiles, which therefore can be used to create maps of cellular processes at a genome-wide scale (*Costanzo et al., 2010*; *Costanzo et al., 2016*; *Fischer et al., 2015*; *Pan et al., 2018*; *Rauscher et al., 2018*; *Tsherniak et al., 2017*; *Wang et al., 2017*; *Yu et al., 2016*).

In addition to univariate phenotypes, such as fitness and growth phenotypes of cells or organisms, genetic interactions can be measured for a broader spectrum of phenotypes by microscopy and image-analysis (*Horn et al., 2011*; *Laufer et al., 2013*; *Roguev et al., 2013*). Importantly, by allowing to infer the direction of specific genetic interactions, multivariate phenotypes further opened the possibility to predict a hierarchy of epistatic relationships of components in genetic networks (*Fischer et al., 2015*).

To date, most studies of genetic interactions focused on 'static' environmental conditions (e.g. under optimal culture conditions), ignoring the impact of context-dependent changes. Recently, several studies have more specifically analyzed the influence of environmental changes on genetic interactions (*Bandyopadhyay et al., 2010*; *Billmann and Boutros, 2017*; *Díaz-Mejía et al., 2018*; *Guénolé et al., 2013*; *Martin et al., 2015*; *St Onge et al., 2007*; *Wong et al., 2015*). For example, *Bandyopadhyay et al. (2010)* defined static, positive and negative differential interactions that vary under changing environmental conditions. (*Billmann and Boutros, 2017*) used extrinsic and intrinsic changes of Wnt signaling in cultured *Drosophila* cells to map differential genetic interactions using a pathway-centric functional readout. These studies demonstrated that widespread changes in genetic interactions occur upon changes in environmental conditions. RNA interference (RNAi) can be used to perturb gene function with high efficiency and specificity to study gene function and map genetic interactions in *Drosophila* tissue cell culture (*Heigwer et al., 2018*).

Upon treatment, for example, with small molecules, genetic interactions change over time due to time-dependent inhibition of components or other changes in the underlying composition of its molecular constituents. To date, little is known about the trajectories genetic interaction networks 'rewire' over time and models for their analysis as well as proof-of-principle data sets are missing. In this study, we devised an experimental and analytical approach to gain insights into higher order (e.g. gene-gene-drug) interactions. To analyze how genetic interactions manifest over time, we used a high-throughput, image-based, multivariate phenotypic readout. By combining combinatorial RNAi with a MEK inhibitor or control treatment, we measured higher order chemo-genetic interactions in *Drosophila* S2 cells to gain new insights into the wiring diagram of the Ras signaling cascade.

Ras signaling is an important oncogenic pathway and Ras and EGFR family proteins are frequently mutated in cancer (*Rodriguez-Viciana et al., 2005*). MEK1/2 (the ortholog of Drosophila *Dsor1*) acts downstream of Ras and phosphorylates ERK1/2 (the ortholog of *Drosophila rl*), which phosphorylates many other proteins (e.g. ETS-family transcription factors [*Friedman et al., 2011*]). The topology of the Ras signaling pathway and its key components are widely conserved between human and *Drosophila* (*Kolch, 2005*; *Perrimon, 1994*; *Wassarman et al., 1995*). In *Drosophila*, the Ras-pathway has been implicated in early embryonic patterning, growth of wing imaginal discs, differentiation of photoreceptors and blood cell proliferation (*Asha et al., 2003*; *Prober and Edgar, 2000*; *Wassarman et al., 1995*).

In this study, we first performed a series of high-throughput image-based genome-wide RNAi screens to identify a set of 168 genes with phenotypic profiles sensitive to MEK inhibition. To construct the differential genetic interaction network, we then created a 168 × 76 double-perturbation matrix and measured the effect of 12,768 gene-gene perturbations under differential time and treatment conditions. These perturbations were characterized by 16 reproducible and non-redundant phenotypic features. Notably, we assessed how each treatment-sensitive interaction changes over time and used this information to construct maps of context-dependent biological modules. Context-dependent interactions mapped the plasticity of Ras signaling and cross-talk to other signaling pathways, such as *Rel* and Stat signaling. Our analyses help to better understand the principles of interaction changes in higher order combinations of genetic perturbations.

## Results

### Time-dependent genetic interactions

Previous studies defined positive differential, negative differential and stable interactions between two genes associated with changes in environmental conditions such as DNA-damage inducing agents (*Bandyopadhyay et al., 2010*; *St Onge et al., 2007*). Positive differential interactions are newly forming under stress conditions and mark resistance or other mechanisms counter-acting the noxious stimulus (e.g. drug treatment). Negative differential interactions, on the contrary, mark connections that are required for homeostasis under normal, unperturbed conditions but are either obsolete or harmful under stress conditions. Within these studies, the wiring diagrams of genetic interaction networks were studied at steady state conditions between two endpoints. The information gained from observations of isolated gene-gene-drug interactions thus missed dynamic responses of differential interactions (*Bandyopadhyay et al., 2010*; *Ideker and Krogan, 2012*; *Mani et al., 2008*; *Martin et al., 2015*).

Based on the observation that the formation of measurable genetic interactions appears to be time dependent (*Figure 1A*), our study aims to extend the previously established framework of differential genetic interactions by adding a time component. Often, when genetic interactions such as a synthetic sick or lethal interaction between two genes are quantified, different interactions-scores (π) are found at different time points (*Figure 1B*). This indicates that, next to a perturbation by external stresses (e.g. chemicals), also time influences the experimental outcome of genetic interaction measurements systematically. We thus extended the theoretical concept of context-dependent interactions by adding a temporal component and distinguished time-dependent from time-independent interactions, treatment sensitive versus treatment insensitive and alleviating (rescuing) from aggravating interactions (*Figure 1C*). By a systematic exploration of the time's influence on stress-sensitive genetic interactions, we can gain an understanding on the mechanisms that change genetic

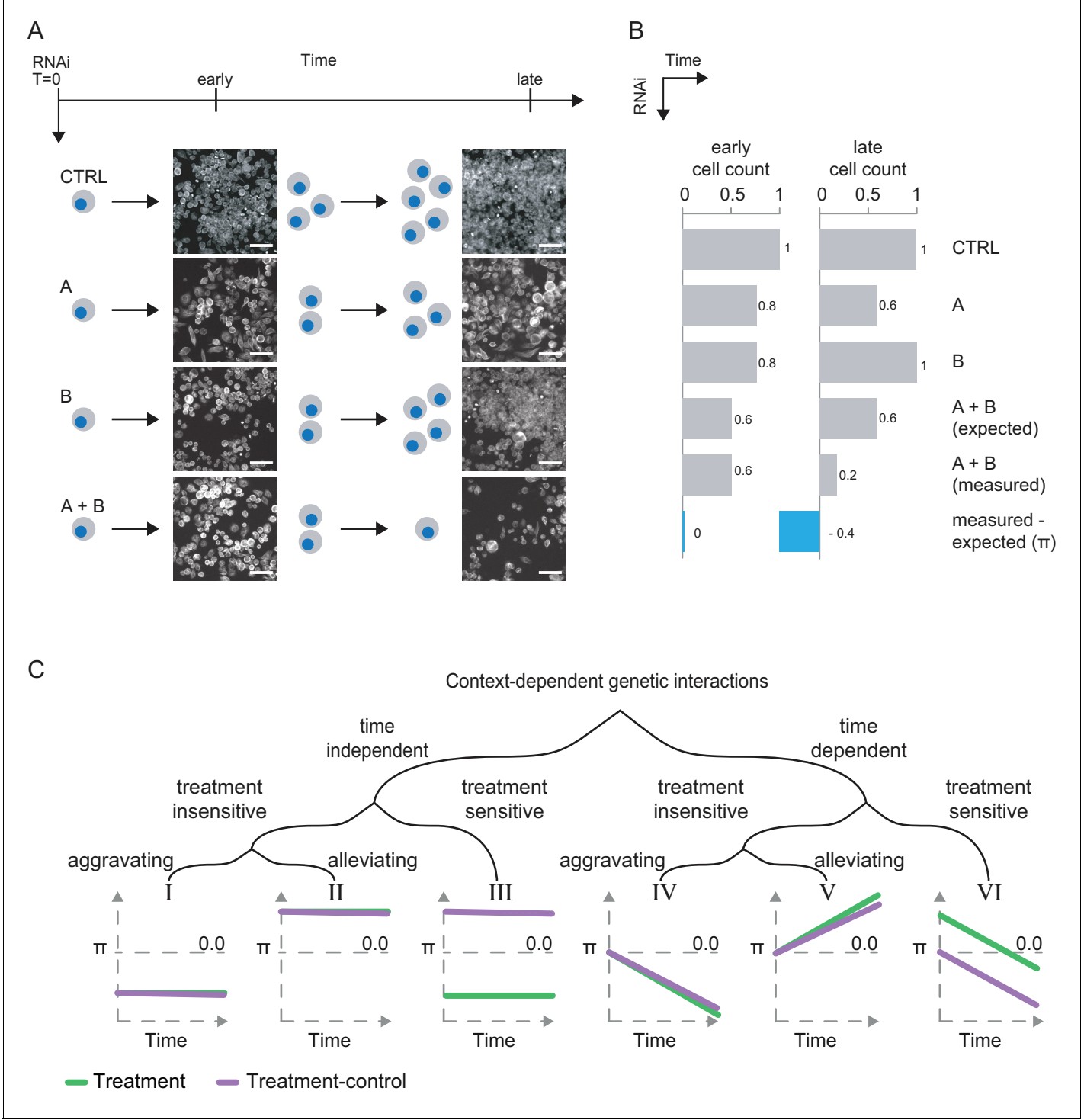

**Figure 1.** Genetic interactions rewire over time. (A) Schematic illustration of a synthetic lethal trajectory between two genes A and B. The co-perturbation of A and B shows no unexpected combinatorial effect at early time points. At later time points, the combined perturbation of both genes prohibits cells from growing and even leads to increased cell death. However, knockdown of A or B alone reduces fitness at either time point. Scale bar is 50 μm. Greyscale image of tubulin (FITC-mAB). CTRL represents non-targeting RNAi. Early = 3 d after dsRNA transfections. Late = 5 d after dsRNA transfection. (B) Interactions can be quantified for each condition by a multiplicative model of interaction as the deviation of the measured combined phenotype from the expected combined phenotype. (C) Theoretical systematic of context-dependent genetic interactions. Interactions can potentially be constant (I-III) or change over time (IV-VI). Interactions can be sensitive (III, VI) or resilient (I–II, IV–V) to an external treatment. Resilient interactions, can be alleviating (II, V, positive π-scores) or aggravating (I, IV, negative π-scores). Sensitive interactions have alternating π-scores (I, III).

DOI: https://doi.org/10.7554/eLife.40174.003

interactions over time, and thus the possibility to map stress responsive interactions in greater depth and the chance to assess the time dependence of stress response of specific biological processes upon chemical perturbation of MEK. Thus, we asked: (i) What is the behavior of genetic interactions over time and how can we describe it? (ii) What do we learn about the genetic interaction network in response to a compound treatment when observed over time? (iii) What specific biological processes underlie time-dependent and treatment-sensitive genetic interactions. (iv) Can we in turn reveal new characteristics of the biological pathways under study, for example regulatory feedback loops in Ras signaling in response to MEK inhibition?

## A chemo-genetic screen identifies genes sensitive to small molecule MEK inhibition

To recover a broad spectrum of cellular phenotypes upon MEK-inhibition, we used a cell morphology assay and automated image analysis in *Drosophila* cells (*Breinig et al., 2015*; *Fischer et al., 2015*; *Horn et al., 2011*). *Willoughby et al. (2013)* previously compared the effect of multiple small molecule MEK inhibitors in vivo and in S2 cell culture and showed that all but one inhibitor significantly reduced the levels of phosphorylated *rl*. In this assay, we perturbed cells by small molecule treatment and genetic perturbagens before we arrested cellular morphology by fixation and stained for DNA (visualizing the nucleus), actin (visualizing cell adhesion and cytoskeleton organization) and α-tubulin (visualizing cell morphology and spindle apparatus). Using automated high-throughput microscopy combined with a real-time image analysis framework we then recorded morphological phenotypes on a single-cell level. The resulting multivariate phenotypic feature vectors describe the quantitative phenotype resulting from the perturbations (*Figure 2*, *Figure 2—figure supplement 1A*, Materials and methods).

As combinatorial gene perturbation screens scale poorly with the number of genes, we first sought to identify genes which phenotypes change in a MEK-inhibitor-sensitive manner. Previous studies have found that genes involved in gene-gene interactions are enriched for genes that themselves display a phenotype distinguishable from the wild type (*Deshpande et al., 2017*; *Koch et al., 2017*). Hence, the identification of genes showing a phenotype as a single knockdown will likely enrich combinatorial screens for genes that form higher order interactions. To this end, we performed multiple genome-wide RNAi screens under different environmental conditions (*Figure 2—figure supplement 1*, Materials and methods, Appendix 1).

For the following gene-gene interaction analysis, we selected a set of 168 genes from the genome-wide screens that showed: (i) high reproducibility between biological replicates, (ii) high correlation between sequence-independent dsRNA reagents (Pearson's correlation coefficient [PCC] > 0.5), (iii) measurable effects that deviate from the negative controls, (iv) differential phenotypes upon *Dsor1* inhibition, and (v) are expressed in S2 cells (log normalized read count > 0, see *Supplementary file 1*). We also prioritized genes that were largely uncharacterized (Materials and methods, Appendix 1). The resulting gene list for gene-gene interaction screening includes 168 target genes that also cover a number of signaling pathways including Ras signaling, innate immunity, Wnt signaling, mRNA splicing, protein translation, cell cycle regulation, Jak/STAT and Tor signaling (see *Supplementary file 2*). The query gene set, a subset of the 168 target genes, contained 76 well-described genes to aid biological interpretability.

## A time resolved co-RNAi screen to capture differential genetic interactions

To quantitatively analyze treatment-sensitive genetic interactions in a time-dependent manner, we set up an experimental design based on co-RNAi treatment and high-throughput microscopy (*Figure 2A*). A combinatorial gene-gene matrix covering 168 target genes and 76 query genes was used to measure 12768 genetic interactions under the different conditions. The library was screened under MEK (Dsor1) inhibitor and control conditions at 48, 72 and 96 hr after compound addition. The screen was performed using two sequence-independent dsRNA design replicates and in two biological replicates for each condition. In total 4.4 Mio. fluorescent images were captured, and 155 image features measured the perturbation effects for every single cell in the experiment (Appendix 1). Following automated image analysis, we transformed the phenotypic features using the generalized logarithm, normalized, centered and scaled them (Materials and methods, Appendix 1). Plates

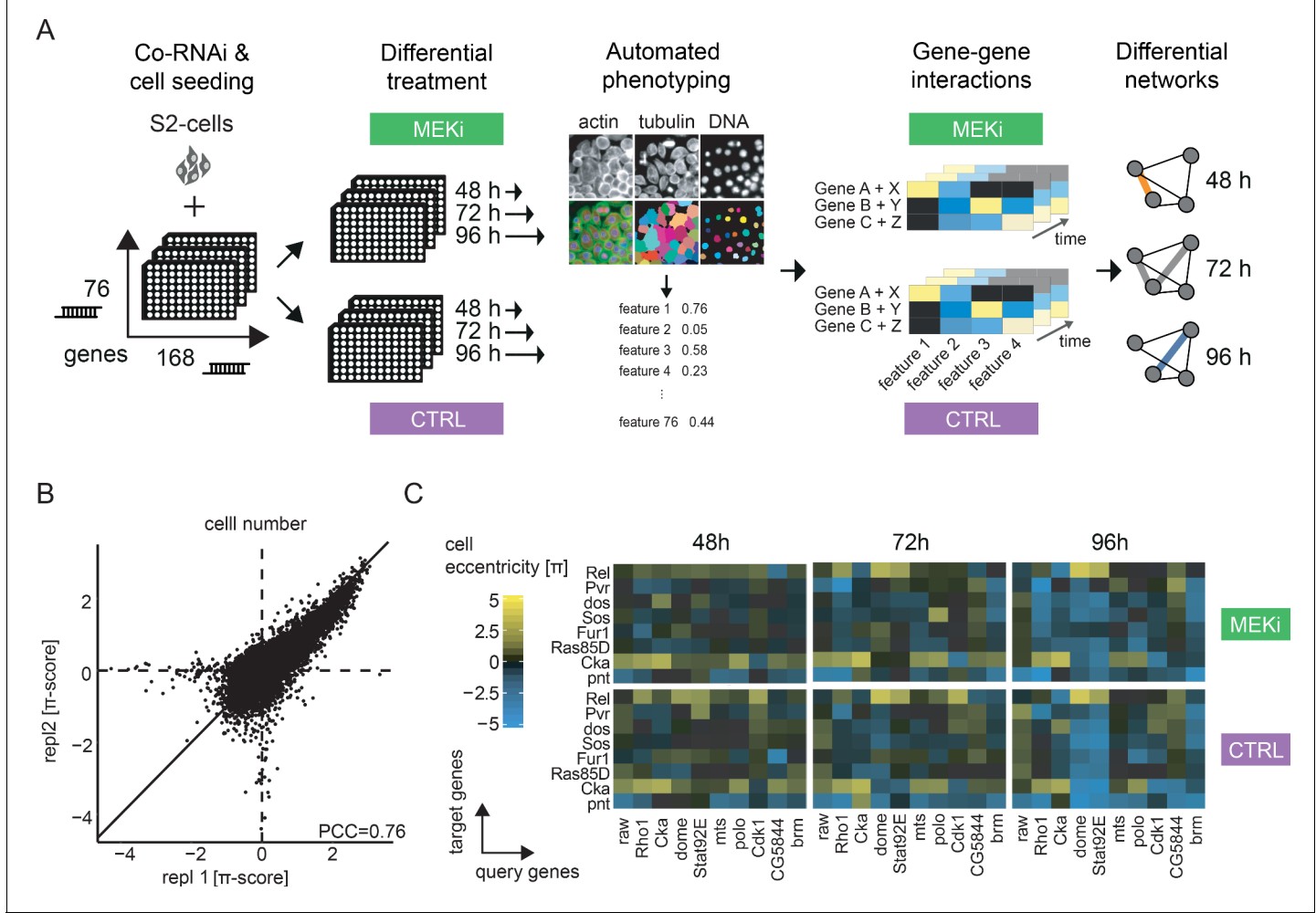

**Figure 2.** An image-based co-RNAi screen maps time resolved genetic interactions. (**A**) Representation of the combinatorial RNAi (co-RNAi) screening setup. 168 'target' and 76 'query' genes were combined to all pairwise combinations and arranged accordingly in 384-well plates. S2 cells were reverse transfected with pre-spotted dsRNAs and incubated for 24 hr. Cells were treated either with small molecule (MEKi [PD-0325901], 1.5 nM) or DMSO (solvent control, 0.5% DMSO) and incubated for additional 48, 72 or 96 hr. The assay was stopped by fixation and staining of cells. Phenotypes were measured using automated microscopy and quantitative image analysis. Genetic interactions (π-scores) were called for 16 non-redundant phenotypic features from the combinatorial knock downs, separately for each treatment and time point. MODIFI was applied to identify significant differential genetic interactions. The model is defined as $\pi_{[A,B,time,treatment]} \sim \sigma_{[A,B]} * time + \delta_{[A,B]} * treatment + \varepsilon_{[A,B]}$ with $\pi$ being the measured interaction for a pair of genes A and B at a given time and treatment. (**B**) Reproducibility of π-scores between biological replicates is high for the exemplary feature 'cell number' (PCC = 0.76). (**C**) Example of genetic interactions observed over time and treatment. Interaction data for the inhibitor treated and control condition are shown for eight selected target genes (y-axis) and 10 query genes (x-axis). Genetic interactions shown were calculated for the cell eccentricity feature.

DOI: https://doi.org/10.7554/eLife.40174.004

The following figure supplements are available for figure 2:

**Figure supplement 1.** Measuring chemo-genetic interactions by high throuput imaging and RNAi.

DOI: https://doi.org/10.7554/eLife.40174.005

**Figure supplement 2.** A large proportion of phenotypic features deliver independent information.

DOI: https://doi.org/10.7554/eLife.40174.006

**Figure supplement 3.** Screening quality control.

DOI: https://doi.org/10.7554/eLife.40174.007

**Figure supplement 4.** Image derived phenotypes closely resemble those of preceeding screens.

DOI: https://doi.org/10.7554/eLife.40174.008

**Figure supplement 5.** Genetic interactions are reproducible, permutation agnostic and non-redundant.

DOI: https://doi.org/10.7554/eLife.40174.009

*Figure 2 continued*

**Figure supplement 6.** Dsor1 inhibiting effect of PD-0325901 shown by loss of rl phosphorylation, loss of viability and differential phenotypic responses.
DOI: https://doi.org/10.7554/eLife.40174.010
**Figure supplement 7.** Ability of the cell morphology assay to distinguish phenotypes produced by control dsRNA.
DOI: https://doi.org/10.7554/eLife.40174.011
**Figure supplement 8.** Measuring chemo-genetic interactions by high-throughput imaging and RNAi.
DOI: https://doi.org/10.7554/eLife.40174.012
**Figure supplement 9.** Full immunoblots.
DOI: https://doi.org/10.7554/eLife.40174.013

failing technical quality control (Z'-factor between *RasGAP1* RNAi and Diap1 RNAi <0.3 and biological correlation <0.6 PCC for cell number) were masked in further analysis. Overall,<3% of all plates were excluded according to these criteria. Most of the 155 features showed a high reproducibility (80% having a PCC greater than 0.6, *Figure 2—figure supplement 2A*). The two features cell count (relative cellular fitness) and actin eccentricity (morphology of cells) were among the features with the highest replicate correlation (*Figure 2—figure supplement 3A,B*) and are highlighted as exemplary features in some of the following visualizations. All features that failed to meet a replicate correlation of PCC >0.6 were removed, leaving 114 features for further analyses. In addition, 90% of sequence-independent dsRNA pairs correlate with a PCC >0.6 with an average correlation of PCC = 0.77 (*Figure 2—figure supplement 3C*).

Since many of the remaining 114 features provide redundant information (*Figure 2—figure supplement 2B*), overlap was reduced by first clustering all features according to the pairwise PCC of the genetic interactions. Second, we fixed the first feature (cell number) and removed all remaining features that correlated with PCC >0.7. Third, we selected the next most reproducible and biologically interpretable feature and removed all highly correlated features; this scheme was iterated until all features were passed. The remaining 16 features (see *Supplementary file 3*) were selected for further analysis. As a confirmation, we verified that cell number and actin eccentricity show a weak correlation (PCC = 0.48) and thus provide independent information (*Figure 2—figure supplement 2C*). An unbiased 'information gain' analysis by stability selection, as carried out in an earlier study (*Fischer et al., 2015*), validated this approach showing that each of the chosen features also delivers independent but reproducible information (*Figure 2—figure supplement 2D*). As they enrich biologically interpretable and reproducibly measurable features, we however kept the features selected by correlation-based analyses. An analysis of the multivariate Z'-factors between RasGAP1, a negative regulator of Ras signaling and Pvr, a positive regulator of Ras signaling (*Zhang et al., 1999*) showed a multi-variate Z' of 0.814, indicating high assay quality (*Figure 2—figure supplement 3D*).

In a first quality control step, we systematically analyzed whether: (i) π-score analysis recapitulates earlier studies using a cell morphology readout in *Drosophila*, (ii) π-scores were reproducible between biological replicates, (iii) the interaction profile changed considerably when target and query genes switch roles and (iv) interaction profiles were independent for different features. To this end, we compared gene-gene interactions that overlapped between this and previous studies of genetic interactions in *Drosophila* S2 cell culture (*Figure 2—figure supplement 4*). We found significant agreement between π-scores measured in various features in the different studies (FDR << 0.1, for linear dependence between π-scores measured in different studies). We found, for example, that the DNA texture feature we used, could also explain the phospho-histone H3 staining used in *Fischer et al. (2015)*.

Next, we confirmed a high correlation of interactions between biological replicates, as illustrated on the phenotypic features 'DNA eccentricity' and 'cell number' (*Figure 2B*, *Figure 2—figure supplement 5A,A'*). As the combinatorial matrix contained all query genes also in the target gene set, we tested whether interaction phenotypes were in accordance regardless of the assignment of target and query. In theory, all interactions should be symmetric, and it should not matter which gene was assigned as target and which as query. However, in practice target and query RNAi reagents were added independently during the experiment which could skew symmetry. Our analysis demonstrated that both combinatorial conditions highly correlate (*Figure 2—figure supplement 5B,B'*, PCC = 0.76 for cell number; PCC = 0.75 for actin eccentricity). We furthermore confirmed that different features provide independent information about genetic interactions as indicated by low

correlation (PCC = −0.21 and 0.04, *Figure 2—figure supplement 5C,C'*). We also confirmed the suitability of our cell-based assay to score compound induced phenotypes without the need to measure its biochemical effect, and determined the $ED_{50}$ of the MEK-inhibitor PD-0325901 on S2 cells (*Figure 2—figure supplement 6*). These experiments demonstrated that S2 cells show a sustained phenotypic response toward PD-0325901. A high correlation (PCC = 0.81) between small molecule and RNAi perturbation of MEK indicates high compound specificity. The most drastic phenotypic changes among a number of features occurred in a concentration window around the drug's $ED_{50}$ (1.5 nM). Thus, we selected a concentration of 1.5 nM PD-0325901 as an optimal condition for the co-RNAi screening experiments. This ranges within an order of magnitude of the $ED_{50}$ known for treatment of mammalian tissue cells cultures (*Ciuffreda et al., 2009*; *Hatzivassiliou et al., 2013*). Under control conditions, phenotype vectors also reliably separated control RNAi treatments (*Ras-Gap1* [*RASAL3*] vs. *drk* [*GRB2*], *Figure 2—figure supplement 7*). In addition, the multi-variate Z' factor is significantly higher than univariate Z' using cell count only (*Zhang et al., 1999*). We also found that the knockdown phenotypes of known Ras pathway components *Dsor1 (MEK1/2)* and *drk* showed a high correlation (PCC = 0.91, *Figure 2—figure supplement 8*). Accordingly, knockdown phenotypes of genes with antagonizing function like the negative regulator of Ras signaling *Ras-GAP1* and *Dsor1* inversely correlate (PCC = −0.78). dsRNA targeting the same gene were also highly reproducibly producing similar phenotypic vectors (e.g. $PCC_{RasGAP1}$ = 0.88). Hierarchical clustering of phenotypic profiles recapitulated known functional relationships of Ras pathway components, whereas regulators of translation show distinct phenotypes. These experiments demonstrated that the morphological assay captures meaningful phenotypes for MEK inhibition, robustly distinguishes controls and groups functionally related genes into clusters of phenotypic similarity.

Following quality control, we calculated genetic interaction scores ($\pi$-scores) for each feature under each condition using a multiplicative model as described previously by Horn *et al.* (*Horn et al., 2011*, Materials and methods). Overall, we analyzed over 1.3 million gene-gene interactions in two conditions, three time points and 16 cellular features. 72922 interactions showed a significant deviation from the expected combinatorial phenotype. Only 9090 (12%) genetic interactions are measured significantly (moderated t-test [limma], FDR < 0.1) for the cell number phenotype underlining the value of the multiparametric analysis.

## Robust linear modeling of differential genetic interactions across multiple features

*Figure 2C* shows an excerpt of the genetic interaction matrices obtained for each treatment and time condition. We found that our analyses recapitulated known genetic interactions. For example, Ras signaling components showed negative interactions with the Jak/STAT pathway (e.g. *Pvr*, *dos* and *Sos* show negative genetic interactions with *dome* and *Stat92E (STAT5B)*, *Baeg et al., 2005*; *Li et al., 2003*; *Xu et al., 2011*). The observed interactions become stronger over the three time points measured, and interactions such as a negative interaction between Ras signaling components and Rho1 are stronger upon MEK inhibitor treatment.

Next, we sought a suitable statistical framework to score significant context-dependent interactions. Previous studies employed different statistical tests that score the significance of interaction differences between endpoint measurements (*B-Score*, *dS-Score*, limma-based moderated t-test, *Bandyopadhyay et al., 2010*; *Bean and Ideker, 2012*; *Billmann and Boutros, 2017*; *Guénolé et al., 2013*). In a pooled genetic interaction screen in human cells, Shen *et al.* used the time dependence of fitness defects to improve statistical power (*Shen et al., 2017*). Thus, we tested whether we can also leverage a time- and treatment-dependent model (Materials and methods, *Figure 3A*) to identify differential genetic interactions more sensitively than time-independent statistical models (*Figure 3B*).

We found that a robust linear model of serial measurements (MODIFI, *Figure 3A*) identifies the most differential interactions (4723 in total, 2.31% of all possible interactions, FDR < 0.1). When using only end-point measurements, the robust statistic (rlm) is more sensitive than the moderated t-test (limma, *Billmann and Boutros, 2017*; *Fischer et al., 2015*; *Laufer et al., 2013*) used in previous studies to score treatment-sensitive interactions and the two-tailed t-tests (*Bandyopadhyay et al., 2010*; *Guénolé et al., 2013*) of each interaction between conditions (1907 vs 874 vs 21 interactions, respectively). We further found that MODIFI increased statistical power, identifying 147% more differential interactions across all features when compared to the best

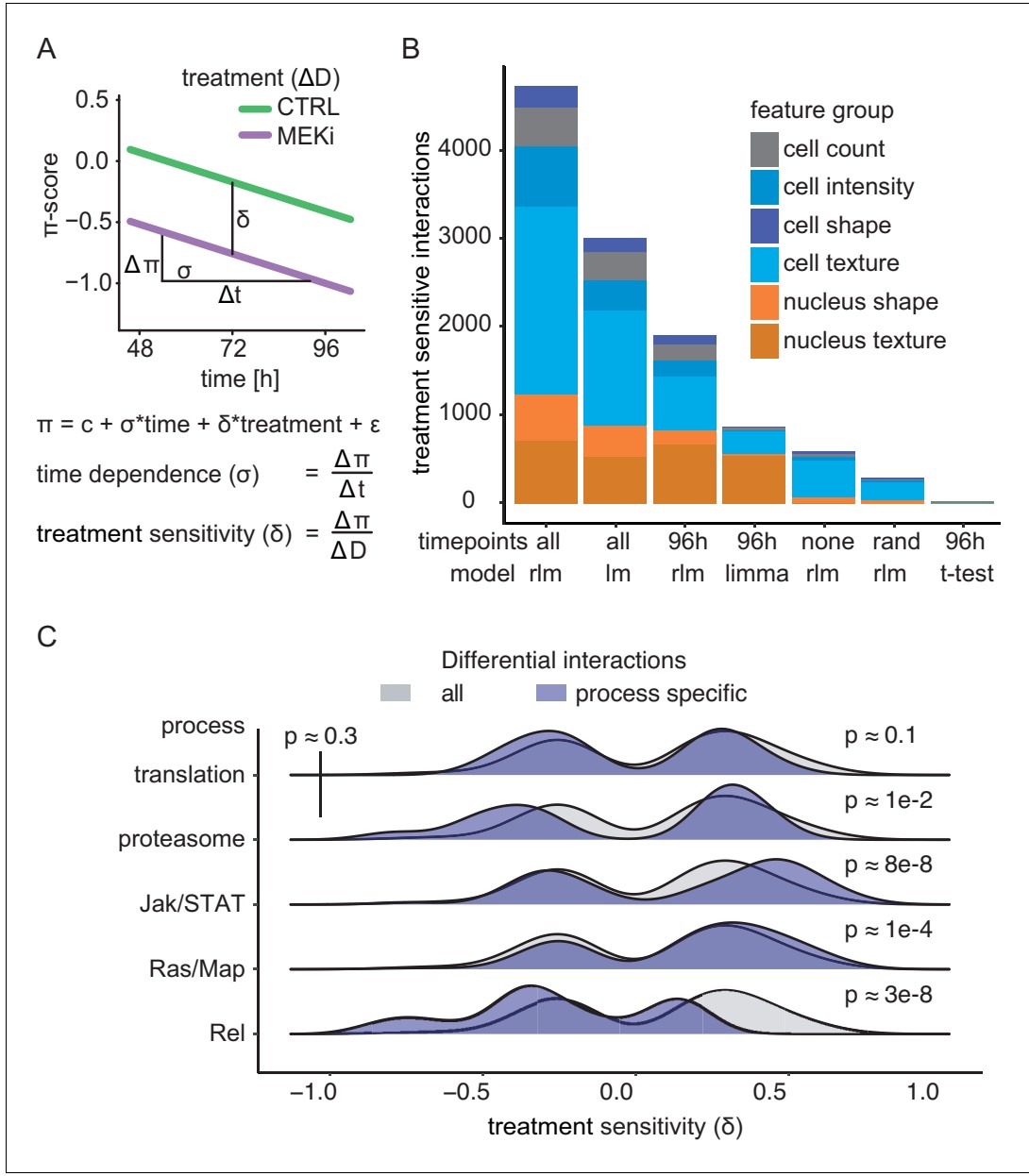

**Figure 3.** Robust linear models describe the continuity of genetic interaction rewiring. (A) Derived measures from the interaction model: the time dependence ($\sigma$) of interaction development, treatment sensitivity ($\delta$) of response to MEKi and average initial interaction difference. (B) Differential interactions detected depending on the used data and model. Significant (FDR < 0.1) differential interactions were counted when analyzed using a linear model (lm), a robust linear model (rlm), a moderated t-test (limma) or Welch's t-test. End-point, sequential and randomized data were compared. The analysis was carried out for all features and accumulated counts are shown. 'none' means that all time points were treated as replicates of the same measurement. 'rand' means that measurements were assigned to random time points and 96 hr denotes the data treated as end-point measurements of the last time point. All models tested the null hypothesis that there is no difference between treatments. A two-sided Welch's t-test was used. The robust linear model (rlm) coefficient's significance was estimated using robust F-tests. The significance of the linear model coefficient was tested by two-way ANOVA. (C) Measurement of sensitivity toward MEKi. The treatment sensitivity was assessed by comparing $\delta$ between biological processes. Significance was tested by a two-sided Kolmogorov-Smirnov test of the sample against all measured interactions. Resulting p-values are indicated. Upper-left p-value compares process specific $\delta$ between translation and proteasome.
DOI: https://doi.org/10.7554/eLife.40174.014

The following figure supplement is available for figure 3:

*Figure 3 continued on next page*

*Figure 3 continued*

**Figure supplement 1.** Robust linear models reliable quantify the differential interactions.
DOI: https://doi.org/10.7554/eLife.40174.015

endpoint measurements (4723 vs 1907; 96 hr/rlm). We conclude that by employing robust statistics MODIFI outperforms conventional models and more accurately estimates the parameters treatment sensitivity δ and the time dependence σ. MODIFI between to genes i an j, is described by the following equation:$\pi_{ij} = c_{ij} + \sigma_{ij} * time + \delta_{ij} * treatment + \varepsilon_{ij}$. There σ estimates the rate by which interactions change and δ estimates the amplitude of interaction change between treatment conditions (Materials and methods).

## Robust linear models accurately describe the temporal dynamics of genetic interactions

We next sought to test if the linear models are practical to describe time and treatment dependent genetic interactions. To this end, we compared the unweighted residuals of each fit with the actual experimental variance measured at each time point. If the model fails to fit the data appropriately (e.g. the comparison does not behave monotonic, or the form of the input-curve is not linear) one would expect that the residuals are unexpectedly greater than the variance. However, analyses of all interactions for each phenotypic feature reveals that this is rarely the case (*Figure 3—figure supplement 1*). In most models, remaining residuals of fit are explained by the variance between biological replicates (avg. PCC = 0.96, $R^2$ = 0.92). Interestingly, this is true for all features we assessed. By initial feature transformation, centering and scaling, systematic differences between features were removed. During subsequent interaction calling, where only the residual of activity not explainable by each single gene knockdown is kept as a phenotype, specific time-dependent behaviors of features affecting each gene are removed as well (*Diss and Lehner, 2018*). We thus concluded that the π-score dependence on time and treatment for each phenotypic feature can be reliably quantified using linear model statistics.

## Related genes share patterns of context-dependent interactions

Next, we quantified to what extent gene-gene interactions changed due to MEK inhibitor treatment (δ). δ serves as a surrogate for the integrated area between the trajectories of the two treatments. If δ is close to zero, only little changes occur upon treatment and high δ marks highly treatment-sensitive interactions. We found that treatment-sensitive interactions were equally likely to be positively or negatively shifted over all analyzed genes (*Figure 3C*, grey distribution). Of note, especially treatment-sensitive interactions of *Rel* (*NFKB1*, a downstream effector of the *Drosophila* Imd signaling pathway, *Myllymäki et al., 2014*), or Ras/Map and Jak/STAT related genes enriched as either negatively shifted π-score or positively shifted π-scores because of MEK inhibition, respectively. This implies that pathways which are positively regulated by MEK tend to form interactions that are less aggravating under MEK inhibition. Interactions formed by *Rel* are negatively enhanced by MEK inhibition. We further found no or little significant difference between housekeeping modules (such as proteasome, translation machinery) and all genes (p>0.1, two-sided KS-test). Taken together, these data suggest that components of the same pathway share differential interaction sensitivity and directionality in response to Ras pathway inhibition.

## Examples of context-dependent genetic interactions

Assessing all interactions for which MODIFI identified statistically significant hits (FDR < 0.1), we identified four main types of time and treatment-dependent interactions that we expected would be recovered by MODIFI (Materials and methods, *Figure 1C* III-VI). Interactions that are neither time-dependent nor treatment-sensitive were not covered by MODIFI (see also *Figure 1C* I, II). Among the time-dependent interactions, we observed alleviating treatment-insensitive interactions where the π-score raised over time (*Figure 4A*). These interactions often involve core essential genes whose influence on the phenotype (e.g. cell count) is not altered by MEK inhibition. This is for example the case for *mts* knockdown (PP2CA, lethal by itself; *Snaith et al., 1996*) where the simultaneous loss of the proteasomal subunit *Prosbeta4* (PSMB2, *Wójcik and DeMartino, 2002*) dominates the

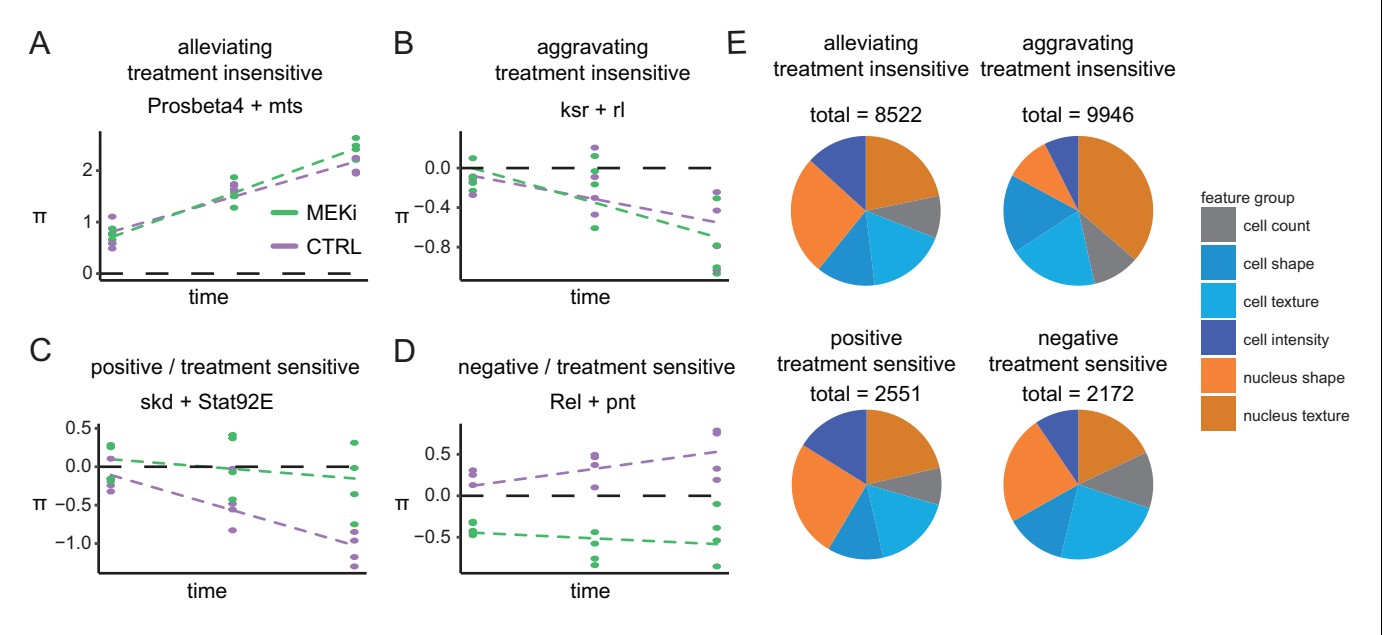

**Figure 4.** Genetic interactions rewire over time. (A–D) Examples of time- and treatment-dependent genetic interactions: (A) alleviating treatment-insensitive interaction of *Prosbeta4* (proteasome) and *mts* (cytoskeleton), treatment invariant and increasing over time, (B) aggravating- treatment-insensitive interaction of *ksr* and *rl* (both Ras signaling), treatment invariant and decreasing over time, (C) positive treatment-sensitive interaction of *skd* (mediator complex) and *Stat92E* (STAT receptor), lifted from synthetic lethal to non-interacting by treatment, (D) negative treatment-sensitive interaction of *Rel* (innate immunity) and *pnt* (Ras signaling), π-scores decreased by the treatment. (cell count, FDR < 0.1, robust f.test followed by multiple testing correction after Benjamini Hochberg). Dashed lines are trendlines for each treatment group. (E) Interaction counts after MODIFI. Interactions (FDR < 0.1) are counted for 16 features, grouped into cell count, shape, texture and intensity within cell and nucleus.
DOI: https://doi.org/10.7554/eLife.40174.016

The following figure supplement is available for figure 4:

**Figure supplement 1.** Time- and treatment-dependent interaction between skd and Stat92E.
DOI: https://doi.org/10.7554/eLife.40174.017

combinatorial phenotype that do not change further regardless of the treatment. In this case, a positive interaction that strengthens over time was measured (*Figure 4A*). Accordingly, we termed interactions aggravating, treatment insensitive when the π-score declined over time and its trajectories were indifferent between treatments (*Figure 4B*).

Aggravating treatment-insensitive interactions on cell count often include signaling transducers where the loss of one only has a mild phenotype while the double perturbation disturbed homeostasis which cannot be buffered buffer and a synthetic sick or lethal interaction is observed. For example, *ksr (KSR1)* and *rl (ERK1/2)*, two core members of the Ras signaling cascade (*Morrison, 2001*; *Wassarman et al., 1995*), interact significantly (p=0.0017). This synthetic sick interaction is stable upon MEK inhibition and thus appears independent of phospho-rl levels which hints toward a kinase-independent function of rl in combination with its scaffolding protein ksr (*Figure 4B*).

We defined interactions as treatment sensitive when trajectories differed significantly between treatments (FDR < 0.1, Materials and methods). If the π-score is lower under control than under treatment conditions, we termed it a positive treatment-sensitive interaction (MEK inhibition lifts the phenotype, *Figure 4C*) and negative treatment-sensitive interaction (MEK inhibition dampens the interaction, *Figure 4D*) in the opposite situation. For instance, *skd* (MED13, an integral component of the mediator complex; *Janody et al., 2003*) showed a positive differential interaction with *Stat92E* (*Drosophila* ortholog of human STAT receptor; *Bina and Zeidler, 2009*) (*Figure 4C*). Under control conditions *skd* knockdown aggravated the fitness loss induced by *Stat92E* knockdown to a lethal phenotype. This aggravation was attenuated under MEK inhibition. Our data suggest that a synthetic lethal relationship connects both genes when they are otherwise unperturbed. Only little is known about the cooperative function of mediator and STAT or crosstalk toward Ras signaling

(*Bina and Zeidler, 2009*). Interestingly, under control conditions, the loss of fitness phenotypes of *Stat92E* and *skd* single knock down are not time dependent, while the interaction is strongly time and treatment dependent. This is indicative of a longer-term transcriptional response when cooperative action of *skd* and *Stat92E* is disturbed (*Figure 4—figure supplement 1*).

In contrast, a negative differential interaction occurred between *Rel* and *pnt*. While *Rel* knockdown rescued the fitness-defect induced by *pnt* knockdown under normal conditions, it aggravated the *pnt* knockdown phenotype after MEK inhibition (*Figure 4D*). Thus, we hypothesize that both, the aggravating interaction between *skd* and *Stat92E* and the alleviating interaction of *Rel* and *pnt* depend on the proper function of Dsor1. Mixed forms, such as interactions that deviate strongly in the beginning experiment and converge later or interactions that were almost time independent but treatment sensitive, were also observed.

## Different phenotypic features capture distinct cellular reactions

To assess whether different features (which we grouped into meta features, such as cell shape or nuclear texture) or pathways show enrichments in one or the other interaction type, we analyzed enrichment of interaction counts over a random distribution. We found considerably more treatment-insensitive than treatment-sensitive interactions for all feature (18468 vs. 4723, 16 phenotypic features, *Figure 4E*). While, as expected, the distribution of negative and positive interactions over all features was symmetric, specific phenotypic features capture surprisingly high numbers of alleviating (nuclear shape) or aggravating (nuclear texture) treatment-insensitive interactions. This indicates that different phenotypic features identify specific biological reactions of cells toward double gene perturbations. A possible explanation would be that different biological processes influence different cellular features, for example perturbations of the cytoskeleton organization mostly influence shape features while perturbation of nuclear factor alters mostly nuclear texture. The direction of interactions then follows the genes that are involved and so do the different features enrich distinct interaction types. Core essential housekeeping genes, for example, show exceptionally high numbers of alleviating interactions on cell shape but simultaneously display mostly aggravated phenotypes on their nuclear texture. These observations indicate a complex interdependence between specific genes under investigation and the phenotypic features that are used to assess them.

## Differential genetic interactions enrich in stress responsive genes and pathways

Additionally, we found that treatment-sensitive interactions, compared to treatment-insensitive interactions, enriched in specific signaling pathways related to MEK inhibition. While, for example, ribosome or spliceosome-related genes formed mostly alleviating and treatment-insensitive interactions (*Figure 5A*), the JNK pathway was enriched for alleviating treatment-insensitive and negative treatment-sensitive interactions (*Figure 5B*). Other pathways, such as Ras signaling, *Rel*, Mediator signaling or Jak/STAT signaling were equally overrepresented in treatment-sensitive and treatment-insensitive interactions. Among the pathways tested, the enrichment of treatment-insensitive interactions highlights pathways with large impact on the interaction network controlling cell viability. The enrichment of differential interactions highlights mainly signaling pathways that are sensitive to MEK inhibition.

Differential genetic interactions are not equally distributed over all genes that were tested. Jak/STAT signaling components (*Stat92E, dome, upd3*) alongside Ras signaling members (*drk, rl, dos, Sos, pnt*) and, interestingly, Imd signaling (*Rel*) showed specific enrichment of differential interactions (cell count feature, *Figure 5C*). Specifically, *pnt* forms many positive differential interactions (alleviated upon MEK inhibition) while *Pvr* is involved in many negative differential interactions (aggravated by MEK inhibition). This could be attributed to *pnt* acting as a terminal transcriptional effector of the signal triggered by the activated receptor *Pvr*. We also found that genes, which form more treatment-insensitive genetic interactions also enrich treatment-sensitive interactions (compare linear trendline, *Figure 5D*). However, some particular genes are involved in unexpectedly many differential interactions. This indicates that a rather specific response to the treatment is reflected in the differential interactions. These data demonstrate that time-dependent modeling of interaction scores sensitively detects treatment differential interactions which enrich in and thus highlight Ras-sensitive biological processes.

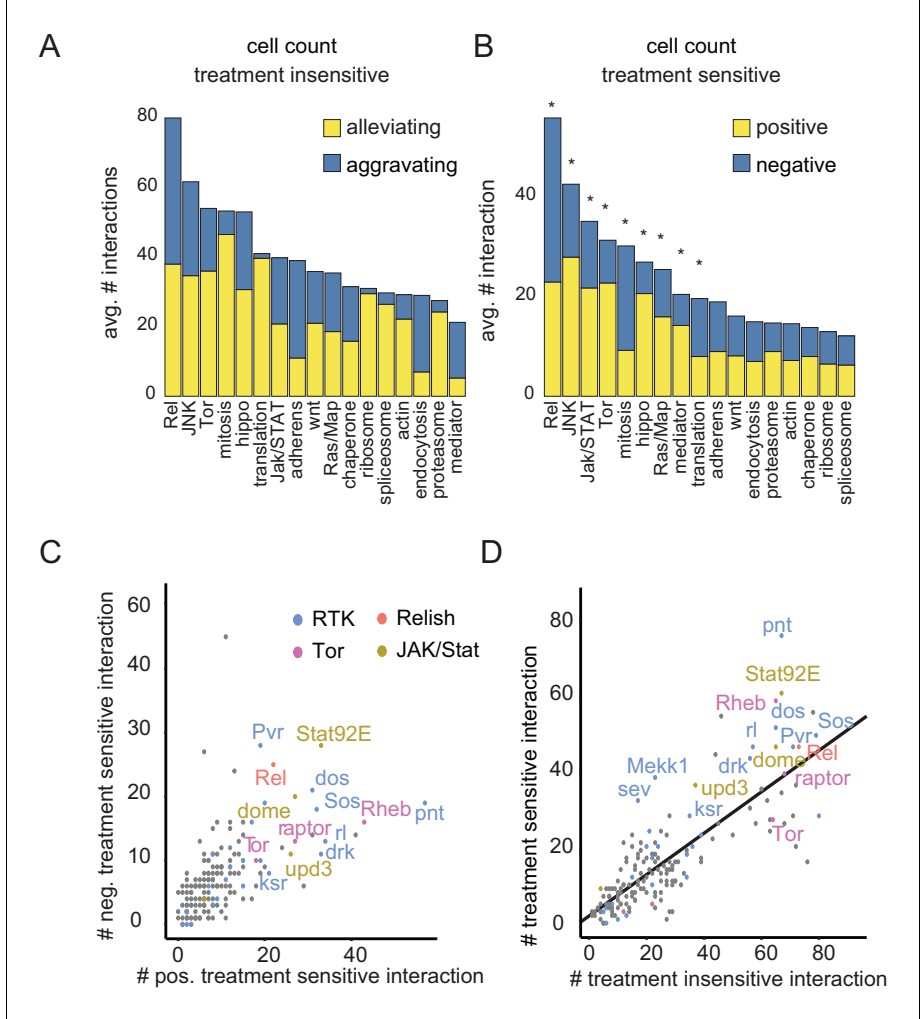

**Figure 5.** Drug sensitive genetic interactions are enriched for stress responsive signaling pathways  (A–B) Distribution of aggravating/alleviating treatment-insensitive (A) and positive/negative treatment-sensitive (B) interactions among molecular pathways. Binomial testing estimated if counts were expected by chance (*=FDR < 0.1). (C) Gene-level interaction counts. Counts of significant, unique negative treatment-sensitive interactions compared to counts of positive treatment-sensitive interactions. Dots are colored by functional groups. Pathways with the most treatment-sensitive interactions (Tor, Ras, Rel and Jak/STAT signaling) are highlighted. (D) Counts of treatment-insensitive interactions are plotted against treatment differential interaction counts. A trendline indicates a general linear dependency between treatment-insensitive and treatment-sensitive interaction counts. (A–D) Count data are based on cell number feature, significant (FDR < 0.3), MODIFI modelled interactions.

DOI: https://doi.org/10.7554/eLife.40174.018

## Signaling pathways rewire with different time dependencies

MODIFI estimates the time dependence ($\sigma$) of each treatment-sensitive interaction. This term can be interpreted as the slope by which an interaction changes (e.g. strengthens or weakens) over time. Depending on the initial difference (compare *Figure 3A*), $\pi$-scores increase or decrease over time, diverge or converge. The most abundant interaction in this study describes a treatment-insensitive interaction that could not be measured initially but forms over the course of the experiment (78% of all significant interactions, FDR < 0.1).

In the following analyses, we use genetic interactions based on cell count as an example to test whether genes or pathways react at different specific rates. For example, from 48 hr to 96 hr after compound addition, genetic interactions with *Rel* remained stable, whereas interactions of Jak/STAT

or Ras signaling-related genes changed significantly over time. Interactions with housekeeping-related genes (proteasomal or ribosomal subunits) show phenotypes of an exceptionally high time dependence (*Figure 6A*). These data indicate that interactions of the different biological processes rewire at different rates after perturbation.

We also hypothesized that the difference of interaction scores and their time dependence could inform about the influence of MEK inhibition on different biological modules or phenotypic features. Cell fitness-based interactions formed by proteasome related genes show the strongest phenotypic differences between treatments at the initial and last measured time point (*Figure 6B*). This suggests that proteasome-related genes are involved in particularly strong treatment-sensitive interactions upon MEK inhibition. These interactions interfere with cell proliferation early on during our

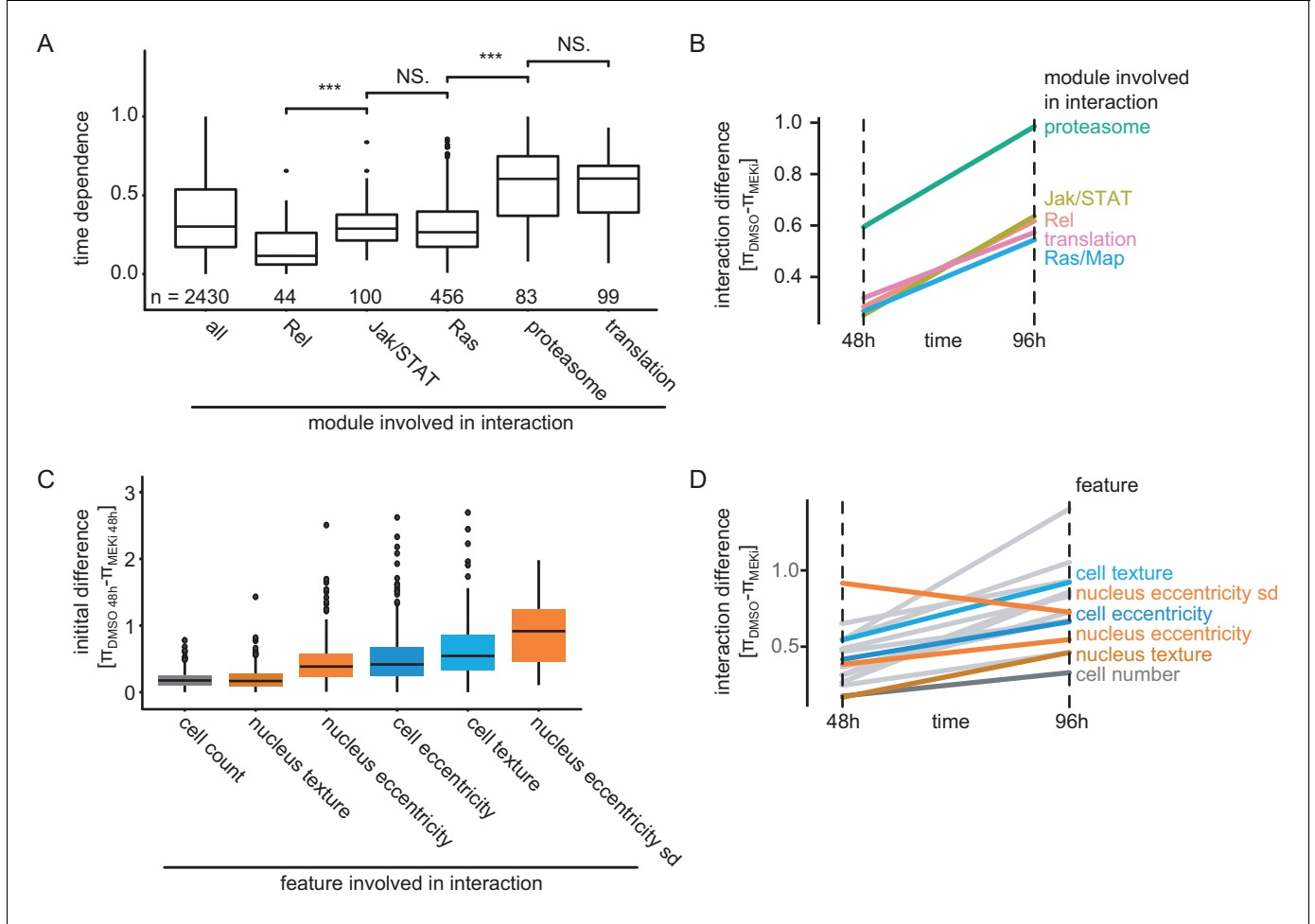

**Figure 6.** Timing and initial difference of interactions depend on the biological process and feature. (**A**) Time dependence (σ) of interactions stratified by biological process. Boxplots show the median (black bar), the 25th and 75th percentile (box) ±1.5 times the interquartile range (whiskers). Points outside that range are plotted individually. Significance is tested by a two-sided welch t-test (***: p<<0.001, NS: p>0.05). Data is shown based on significant (FDR < 0.1) cell-count-based interactions involving genes belonging to this process. (**B**) Median π-score differences stratified by pathway annotation of affected genes. All significant (FDR < 0.1) time-dependent interactions based on cell count feature are summarized by median. Interactions formed by genes that are proteasome associated show the highest initial difference and steepest increase over time. (**C**) Initial difference of interaction scores 48 hr after treatment stratified by feature. Boxplots show the median (black bar), the 25th and 75th percentile (box) ±1.5 times the interquartile range (whiskers). Points outside that range are plotted individually. All features (except nucleus texture) show significantly (p<<0.001, two-sided student's t-test) higher initial differences than cell count based interactions. (**D**) Median π-score differences for the first and the last measured time point. Trajectories for all features are shown over all genes that showed a significantly time-dependent interaction (FDR < 0.1). Features are highlighted by their feature group. All features except nucleus eccentricity measure interaction differences that become more profound over time.
DOI: https://doi.org/10.7554/eLife.40174.019

experiment and also become stronger over time. This supports reports of synergistic effects between proteasome and MEK inhibition on perturbing cell viability (*Chang-Yew Leow et al., 2013*).

Next, we hypothesized that phenotypic features measure different initial interaction differences and analyzed initial π-score differences between phenotypic features. Especially, cell morphology features (nucleus/cell eccentricity) and their variance within the population of cells show initial differences that are significantly higher than those measured by cell count (p<0.0001, *Figure 6C*). Of note, nuclear eccentricity and its variance among the population of cells (nucleus eccentricity sd) are also the only initially different features that are masked later on. All other phenotypes show an increased interaction difference over time (*Figure 6D*). Surprisingly, cell count as the traditional readout for fitness after gene-gene perturbation shows the smallest interaction differences between the treatments in general, irrespective of the time point. Together, these analyses demonstrate that the time dependence of genetic interactions is specific to certain biological process. It further highlights that phenotypes beyond cell viability excel to capture early treatment-sensitive interactions.

## A correlation network of treatment-sensitive interactions maps genes into functional modules

Next, we analyzed whether interaction networks formed by different biological modules or core signaling pathways change systematically over time and treatment. In the following examples, we used cell eccentricity as an exemplary feature which we found to capture early cellular responses. *Figure 7A* shows how an interaction sub-network including Jak/STAT signaling, Ras/Map signaling components and spliceosome related genes rewires over time in reaction to MEK inhibition. Core housekeeping modules (ribosome, spliceosome or proteasome) were highly interconnected by alleviating treatment-insensitive interactions. In contrast, components of the Ras signaling, Jak/STAT signaling or Tor signaling cascade showed aggravating interactions with housekeeping modules. We observed that (i) alleviating interactions (π > 0) dominate early time points, (ii) many initially alleviating interactions reverse over time (π > 0 → π < 0), (iii) differences attributed to the compound treatment become more profound over time. Lastly, we noted that genes in proximity tend to have similar interaction patterns coherently changing over time and treatment (*Figure 7—source data 1*). Previous studies implied that similarities of treatment sensitive genetic interaction profiles can identify functionally related genes (*Bean and Ideker, 2012*). Thus, interactions of related genes change coherently upon network perturbation. Hence, we defined treatment-insensitive interaction profiles for each target gene. We used the modeled interaction difference between treatments over time (δ) to quantify interaction change due to Dsor1 inhibition. For every target, we calculated δ with every query gene in a vector comprising 76 measurements for cell eccentricity.

Correlations between profiles (*Figure 7B*) confirmed known functional relationships of genes, as for example the profiles of the genes *Stat92E* and *dome*, members of the *Drosophila* Jak/STAT pathway, were similar (PCC 0.73) confirming that both genes share biological function upon perturbation of Ras signaling (*Xu et al., 2011*). Furthermore, our analysis showed a correlation of treatment-sensitive genetic interactions for all features between *Stat92E*, *dome* and Ras signaling. Interestingly, the profile of *Rel* was similar to negative regulators of Ras signaling (*RasGAP1*, PCC 0.38), but was anti correlated with positive regulators (*pnt*, PCC −0.37) indicating a potential crosstalk between the two pathways.

We expected that a correlation-based network drawn from treatment-sensitive interaction profiles across all phenotypic features reveals modules of functionally related genes. Thus, we calculated the pairwise correlation coefficients (PCC) of treatment-sensitive interaction profiles (interactions with 76 query genes) including all 16 cellular features of all 176 target genes. We visualized resulting positive correlations in a network graph highlighting biological processes and candidate genes (*Figure 7C*, *Supplementary file 2*). This revealed that correlations of treatment-sensitive interaction profiles clustered genes into known pathway modules. Of note, *Rel* and *Fur1 (FURIN)* and *swm (RBM26)* showed unexpected correlations with members of the Ras signaling cascade (*Figure 7B*).

It is expected that genes with similar functions irrespective of the treatment show similar interaction profiles between and within conditions. In contrast, genes with a treatment-dependent function should lose or gain correlations to other genes when compared between treatments (*Billmann and Boutros, 2017*). To test this, we defined profiles of all interactions across all cellular features and time points and correlated them between genes and between conditions. Most interaction profile

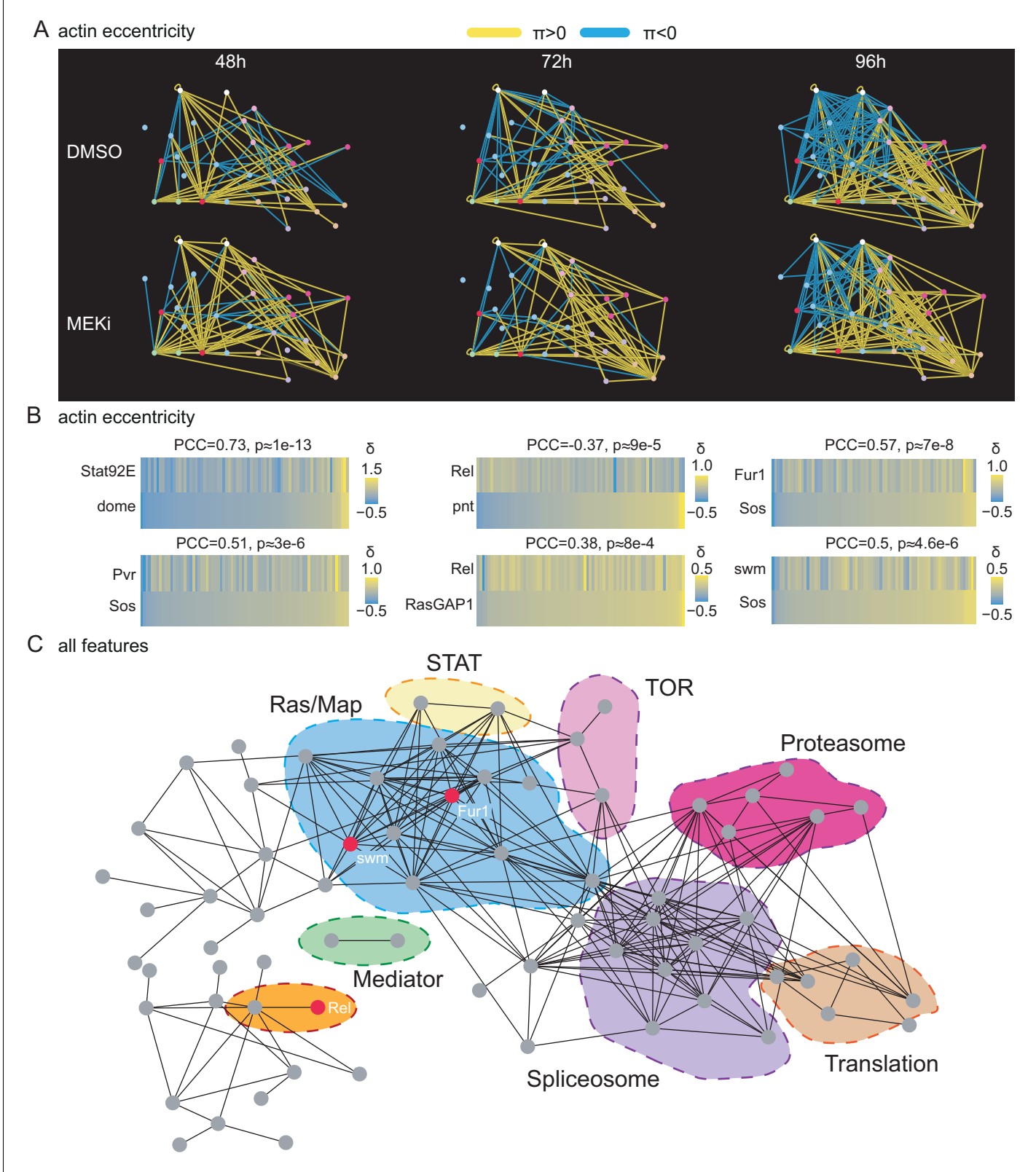

**Figure 7.** A correlation network of treatment-sensitive interactions maps pathways modules. (**A**) Network of genetic interactions between selected genes. The networks include all candidate Ras-signaling (blue), Jak/STAT-signaling (white), Tor-signaling (pink), proteasome (red), translation (orange) and splicing (purple) related genes. Significant (FDR < 0.1) alleviating interactions are shown in yellow. Significant (FDR < 0.1) aggravating interactions

*Figure 7 continued on next page*

*Figure 7 continued*
are shown in blue. All interactions are based on the cell eccentricity feature. Interactions become more abundant and stronger over time, more alleviating interactions can be observed under MEK inhibition. Ras and Jak/STAT related genes are mostly connected by aggravating interactions. (B) Correlations of treatment-sensitive interaction profiles between known regulators and candidate genes. Profiles of all δ-scores along cell eccentricity and 76 query genes were constructed for 168 target genes and pairwise correlations were calculated. Shown is the Pearson correlation coefficient (PCC) and asymptotic p-value as implemented in the R package Hmisc. All correlations shown are significant with an p-value < 0.01. Jak/STAT and Ras components show high correlations as expected. *Rel* appears as negative, *Fur1* and *swm* as positive regulators of Ras signaling. (C) Pairwise correlation network of treatment-sensitive interaction profiles across all 16 features. Shown are all genes with at least one edge. Edges are drawn if two gene's δ-profiles correlate with PCC > 0.5. Nodes are ordered by force directed spring embedded layout. A high degree of clustering of known pathways indicates meaningful correlations.
DOI: https://doi.org/10.7554/eLife.40174.020

The following source data and figure supplement are available for figure 7:

**Source data 1.** Cytoscape session file to reproduce *Figure 7A*.
DOI: https://doi.org/10.7554/eLife.40174.021
**Source data 2.** Cytoscape session file to reproduce *Figure 7C*.
DOI: https://doi.org/10.7554/eLife.40174.022
**Figure supplement 1.** Differential correlation analysis reveals that Fur1 and swm rewire coherently to Ras signaling components.
DOI: https://doi.org/10.7554/eLife.40174.023

correlations did not differ significantly between conditions, compared to within conditions (*Figure 7—figure supplement 1*). Specifically affected gene pairs were mostly Ras signaling components. Interestingly, also profile correlations of Jak/STAT signaling components (*Stat92E*, *dome*) as well as of the two genes *Fur1* and *swm* differed between and within conditions. This provides further clues that *Fur1* and *swm* are implicated Ras signaling. Only few, weak interaction profile correlations were higher between than within conditions.

## Rel and pnt act in a MEK-dependent negative feedback loop

We have shown that the treatment-sensitive interaction profiles of *Rel* and pnt were negatively correlated, whereas *Rel* profiles were positively correlated with *RasGAP1*, a negative regulator of Ras (*Figure 7B*). This suggested that *Rel* itself might function as a negative regulator of Ras signaling. We observed that *Rel* depletion alone had little impact on cell growth, as compared to *pnt*, but showed a cell length (major axis) phenotype (*Figure 8A*, *Figure 8—figure supplement 1*). Co-depletion of *pnt* and *Rel* altered both cell number and cell length. Under control conditions, depletion of *Rel* alleviated the loss of viability and cell length phenotypes after *pnt* knockdown (*Figure 8A*). This interaction was attenuated under MEK inhibition (*Figure 8B*) when co-depletion of *Rel* and *pnt* led to a synthetic lethal phenotype (FDR < 0.1, *Figure 8C,C'*). These interactions were observed for both dsRNA designs (PCC = 0.88 and 0.96 for *Rel* and *pnt*, *Figure 8—figure supplement 2*).

*Pvf2* (orthologue of human VEGF) is upregulated in the absence of *Rel* (log$_2$fold-change = 1.5) (*Boutros et al., 2002*). The data presented here indicate that a knockdown of *Rel* induced a re-activation of the Ras pathway which is dependent on Dsor1 activity (*Figure 8A–C*). We hypothesized that *Rel* negatively regulates Ras signaling by repressing the expression of *Pvf2*, the ligand activating the Pvr-Ras-phl-Dsor1-rl-pnt signaling cascade after binding to Pvr (PDGFR). To test this hypothesis, we performed qPCR analysis of *pnt*, *Rel*, *Pvf2*, *sty (SPRY2)* and *RasGAP1* expression levels (*Figure 8D,E*). We first confirmed the upregulation of *Rel* after depletion of Ras (*Figure 8D*) and showed that upregulation of *Rel* was suppressed by *pnt* co-RNAi. *Pvr* knockdown, as a control for loss-of-Ras signaling activity, led to a downregulation of *pnt* and *RasGAP1*. *Pvr* knockdown also induced a strong upregulation of *Rel* expression. Finally, co-RNAi of *Rel* and *pnt* induced a significant increase in *Pvf2* expression, not observed by depletion of either gene alone (*Figure 8E*). The *Rel/pnt* co-RNAi also induced upregulation of negative regulators of Ras signaling *sprouty* (*sty*) (*Casci et al., 1999*) and *RasGAP1* (*Feldmann et al., 1999*) (*Figure 8E*), thereby providing a mechanistic explanation how *Rel* could negatively regulate Ras signaling.

We hypothesized that this regulatory loop is mediated by the transcriptional regulation of *Pvf2* and requires Dsor1-mediated Ras signaling activity, as summarized in *Figure 8F*. These changes were observed both at 48 hr and 96 hr time-points (*Figure 8—figure supplements 3* and *4*). Interestingly, protein levels of rl were down regulated by *pnt*-or *rl*-RNAi and rescued by *Rel* co-RNAi

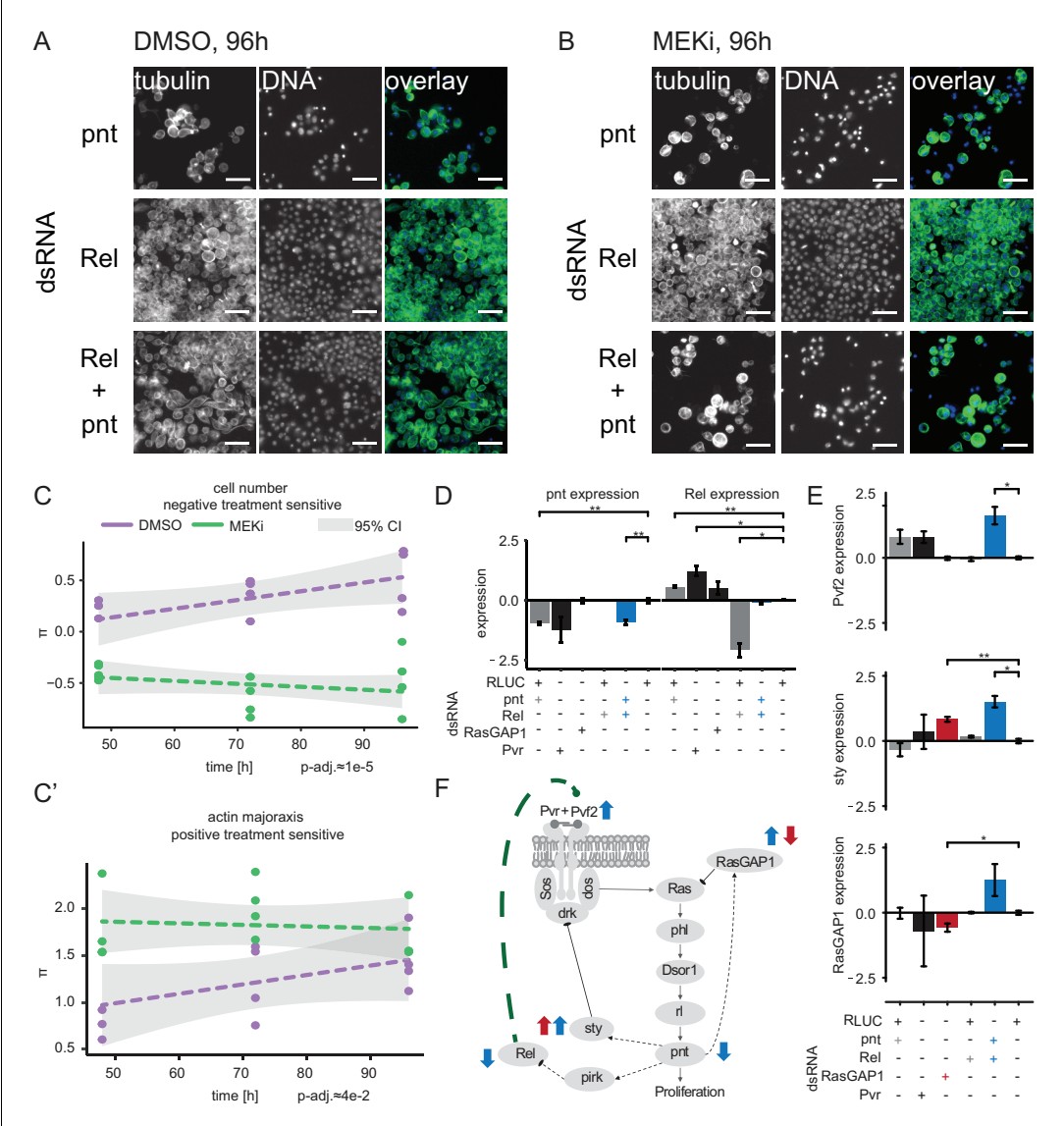

**Figure 8.** Crosstalk of NF-κB and Ras signaling through *Rel* and *pnt*. (A) Upon *Rel* or *Rel/pnt* knockdown cells behave normally, growth is inhibited upon *pnt* knockdown alone. (B) Dsor1 inhibitor treatment attenuated this alleviating interaction. Scale bar = 30 μm. Images are pseudo colored, DNA/DAPI = blue, FITC/α-tubulin = green. (C) Quantified negative treatment-sensitive interaction between *Rel* and *pnt*. The trajectory of the MEK inhibitor treatment is lower than the solvent control condition for cell count interaction indicating synthetic lethality under MEK inhibition. (C') Actin major axis shows a strong positive interaction (cells are enlarged like under *pnt* knockdown). Error of fit is shown as 95% confidence interval. Dashed lines show trendlines of a treatment wise model fit. (D, E) Expression of candidate and marker genes assessed by qPCR (3 days RNAi treatment, n = 3, $\log_2$ fold$_{RLUC}$, mean ±s.e.m., t-test) on S2 cells. (D) *pnt* expression is reduced upon *pnt* and *Pvr* knockdown. *Rel* knockdown does not rescue *pnt* expression. *Rel* expression is increased upon *pnt* and *Pvr* knockdown and decreased upon *Rel* knockdown. Upon *pnt* and *Rel* knock down, *Rel* expression is rescued to normal levels. (E) Pvf2 expression is induced only upon *Rel/pnt* double knockdown. This leads to increased expression of sty and *RasGAP1*. *RasGAP1* knockdown increases sty expression and decreases *RasGAP1* expression. (D–E) *=p < 0.05, **=p < 0.01. (F) A model summarizes the qPCR results in context of the Ras signaling cascade. Dashed lines are transcriptional interactions, solid lines are protein-protein interactions. All black interactions are known, while the green interaction is inferred from the data. Blue arrows indicate that *Pvf2*, *sty* and *RasGAP1* were upregulated upon *Rel/pnt* co-knockdown and by that Ras pathway activity was restored. A similar pattern could be observed upon *RasGAP1* knockdown, which causes intrinsic hyper-activation of Ras signaling by constitutive Ras activation (measured by upregulation of *sty*, red arrows).

DOI: https://doi.org/10.7554/eLife.40174.024

The following figure supplements are available for figure 8:

**Figure supplement 1.** Genetic interactions were summarized under the multiplicative model.

DOI: https://doi.org/10.7554/eLife.40174.025

**Figure supplement 2.** Rel and pnt show high phenotypic agreement between sequence-independent dsRNA designs.

*Figure 8 continued on next page*

*Figure 8 continued*

DOI: https://doi.org/10.7554/eLife.40174.026

**Figure supplement 3.** pnt knockdown strongly induces rl phosphorylation and reduces rl levels after 96 hr.

DOI: https://doi.org/10.7554/eLife.40174.027

**Figure supplement 4.** Instead of phosphorylation, total expression of rolled is altered by Relish knockdown.

DOI: https://doi.org/10.7554/eLife.40174.028

**Figure supplement 5.** Full immunoblots.

DOI: https://doi.org/10.7554/eLife.40174.029

**Figure supplement 6.** Full immunoblots.

DOI: https://doi.org/10.7554/eLife.40174.030

(*Figure 8—figure supplement 4F,G*). Overall, these experiments provide a mechanistic basis how *Rel* acts as a negative regulator of Ras signaling in a context-dependent manner.

## Discussion

To better understand context-dependent differences in genetic networks upon changes in environmental conditions is a current frontier in genetics (*Rancati et al., 2018*). Many biological processes rely on context-dependent changes in genetic requirements, from robustness of cell differentiation during development to responses of cancer cells to chemotherapeutic treatments. However, only few studies on selected phenotypes have systematically analyzed how environmental changes impact genetic interaction networks. Previous studies have analyzed genetic networks after activation of the DNA damage response signaling in yeast or changes in Wnt signaling activity in *Drosophila* cells (*Bandyopadhyay et al., 2010*; *Billmann and Boutros, 2017*; *Díaz-Mejía et al., 2018*). In these studies, positive and negative treatment-sensitive, and treatment-insensitive interactions have been determined based on fitness phenotypes or pathway reporter activity in static end-point assays. Aim of the present study was to analyze changes in genetic networks that impact a broad spectrum of phenotypes by imaging and multiparametric image analysis and to determine how treatment-sensitive interactions change over time after small molecule perturbation of the Ras signaling pathway.

In this study, we established a high-throughput image-based assay which enabled us to reproducibly measure many phenotypes including cell proliferation and cell morphology which are influenced by many cellular processes (*Breinig et al., 2015*; *Fuchs et al., 2010*; *Horn et al., 2011*). We used this assay to measure genetic interactions between differential treatment conditions over the course of three time points. To this end, we assessed the phenotypes of 76608 di-genic interactions in *Drosophila* hemocyte-like cells. Each interaction was characterized by a vector of 16 non-redundant and quantitatively reproducible phenotypic features. Further, we developed MODIFI, a two-factor robust linear model to quantitatively describe the time and treatment-dependent changes of genetic interactions. MODIFI also allowed us to describe whether an interaction is treatment sensitive (treatment could predict π-score) or time dependent (time predicted the π-score). Using MODIFI we found, for example, treatment-insensitive interactions within the Ras signaling cascade (*rl-ksr* interaction, *Figure 4B*) as well as treatment-sensitive crosstalk between Mediator, STAT and Ras signaling (*Figure 4C*). Discovery of such interactions can lead to new treatment options in cases where the pharmacologic inhibition of MEK had no effect and an inhibition of the ERK (*rl*) -KSR (*ksr*) interaction becomes an interesting target (*Roy et al., 2002*; *Yu et al., 1998*). Regarding the example of a signaling axis between Ras, STAT and Mediator signaling, some evidence indicates that mediator and STAT signaling engage in cooperative transcriptional regulation dependent on the phosphorylation status of Stat92E (*Kuuluvainen et al., 2014*; *Wienerroither et al., 2015*). Additionally, evidence exists on mutual crosstalk between phosphorylation-dependent Ras signaling and *Stat92E* (*Li et al., 2002*). Hence, our data suggests that these pathways could be interconnected and *Stat92E* and *Mediator* only show cooperative action when Ras signaling is active and Stat92E phosphorylation is not impaired.

Our analysis showed that we detected treatment sensitive interactions more sensitively as compared to endpoint measurements or single time point replicates (see examples in *Figure 4C,D* & *Figure 8C,C'*). Enrichment of treatment-sensitive interactions among stress-responsive pathways and genes underlines their biological relevance. Using this approach, we also analyzed the treatment (δ)

and time (σ) dependency of interactions of specific genes and pathways. Overall, measuring phenotypes resulting from genetic interactions increased our ability to detect treatment-sensitive interactions. Furthermore, the measurement of multiple phenotypic features simultaneously enabled more detailed characterization of the observed treatment-sensitive interaction. We also tested whether the establishment of phenotypes is dependent on a gene's expression level but found no correlation of high gene expression and high time dependency (data not shown). Our data further suggests that σ is influenced by the general resilience of a pathway or signaling module to perturbations. For example, housekeeping genes, widely believed to form extremely time and condition stable regulatory networks took the most time to rewire their interactions. This makes it unlikely that the stability or turnover of a single gene product is a major driver of time-dependent establishment of genetic interactions. We found, for example, that genetic interactions of 'core' (or housekeeping) modules such as the translation machinery, proteasome and others induce phenotypes that are much stronger at later time-points. In contrast, other cellular modules such as signaling and innate immunity 'rewire' early in the experiment. We could also show that while single perturbation phenotypes in some instances do not change over time interactions still do, hinting toward a time-dependent combinatorial effect. Our analysis classified genes into categories of genetic interactions that are (i) signaling modules central to the cells' physiological role, (ii) signaling modules required for maintaining homeostasis and (iii) resilient 'core' modules whose network hubs form interactions on a longer timescale. We also found that measuring different phenotypes provided more information about the development of interaction differences over varying time scales and demonstrate a number of example treatment-sensitive interactions that could not have been found in end-point assays. While the cell count (comparable to yeast colony size) as a phenotype captures cellular reactions rather late in the experiment, other phenotypes, such as nuclear morphology or cytoskeleton texture, enabled to measure immediate cellular reactions.

In the gene-drug interaction experiments, we found that pathways interacting with Ras signaling reacted strongest to the Dsor1/MEK inhibition. This observation was translated to map signaling modules that react similarly toward Ras signal perturbation and we correlated δ-profiles along all features between all target genes. By this means, genes whose interactions change coherently upon Dsor1 inhibition are grouped into highly interconnected modules. Consequently, this correlation network clusters genes of similar functions in proximity with each other. Each module is also characterized by a coherent reaction towards Dsor1 perturbation. Interestingly, *rolled (rl)*, *Dsor1* and *pole hole (phl)* (ERK1/2, MEK1/2 and Raf) were not connected to the rest of Ras-signaling-related genes in the correlation network. In contrast, they correlated with Ras when using interaction profiles of the control treatment. This indicates that the chemico-genetic analysis identified 'responsive' factors that can be uncoupled upon environmental modulation of specific signaling modules.

Our analysis also revealed three genes that unexpectedly connected to Ras signaling: *Fur1*, a serine-type endopeptidase (*Kim et al., 2015*), *swm*, involved in mitotic checkpoint regulation and hedgehog signaling (*Casso et al., 2008*; *Dong et al., 1997*) and *Rel* (*Foley and O'Farrell, 2004*). The correlation of *Fur1* and *swm* with positive regulators of Ras signaling indicates that they respond similarly towards Dsor1 inhibition as Ras pathway members. In addition, we identified *Rel (NF-κB)* as a strong treatment-sensitive genetic interactor, suggesting that mitogenic Ras signaling and innate immune pathways depend on each other. Once *Rel* is lost, cells become more dependent on Ras signaling; a phenotype that can be blocked by perturbing Dsor1 activity chemically or genetically. Already at a low dose, both perturbations result in a synthetic lethal phenotype that kills *Drosophila* hemocyte-like cells. Conversely, it was previously shown that Ras signaling influences *Rel* activity by regulation of its negative transcriptional regulator pirk (*Ragab et al., 2011*). We hypothesize that this mutual negative feedback regulation could be the basis for a 'fight' or 'flight' response of the immune cells; balancing an immune and proliferative response in the same cell.

Large-scale studies on gene essentiality have challenged the concept of a static repertoire of essential genes. In contrast, loss-of-function screens in different genetic background of cancer cells identified 'core' and 'genotype'-dependent sets of essential genes. This indicates that essentiality is modulated in a context-dependent manner (*Hart et al., 2015*; *McDonald et al., 2017*; *Rauscher et al., 2018*). At this point, our study is the largest exploration of gene-gene-drug interactions based on multiparametric, non-essential phenotypes. We demonstrate how different vulnerabilities for a diverse set of automatically scored phenotypes change upon time and environmental conditions. Our modeling approach increases the confidence to call treatment sensitive interactions

upon changes of environmental conditions. This allows to map a correlation network of cellular modules that react coherently toward the external stimulus. We expect that, when further studies of context-dependent genetic interactions will become available, a comparative analysis will provide fundamental insights into how different cellular networks react to environmental stimuli with implications for therapy resistance and timing of drug treatments. In future studies, MODIFI could be further expanded to include terms assessing the influence of the treatment on the behavior over time (rate of interaction change). This will, for example, aid to understand the qualitative relationships between different treatment trajectories beyond the current analysis and sets the basis for further experiments. This study introduced an experimental and analysis framework to explore time-dependent rewiring of genetic networks which can be used to dissect the complexity of biological networks in model organisms and human cells.

# Materials and methods

**Key resources table**

| Reagent type (species) or resource | Designation | Source or reference | Identifiers | Additional information |
|---|---|---|---|---|
| Cell line (*D. melanogaster*) | Dmel-2; S2 cells; Schneider S2 cells | ATCC CRL1963 from ThermoFisher, Waltham, MA, used in PMID:21378980; PMID: 26912791; PMID:25748138; | ATCC Cat# CRL-1963; RRID: CVCL_Z232 | |
| Antibody | alpha-tubulin-FITC mAB, tubulin antibody, mouse monoclonal clone DM1A | Sigma-Aldrich Cat# F2168 | Sigma-Aldrich Cat# F2168, RRID: AB_476967 | (1:1500) |
| Peptide, recombinant protein | Phalloidin-TRITC antibody | Sigma-Aldrich Cat# P1951 | Sigma-Aldrich Cat# P1951, RRID: AB_2315148 | (1:6000), TRITC conjugate, $C_{62}H_{72}N_{12}O_{12}S_4$ |
| Chemical compound, drug | Hoechst 33342 | Thermo Fisher Scientific Cat# H1399 | Thermo Fisher Scientific Cat# H1399 | (1:4000) from 1 mg/ml |
| Chemical compound, drug | DMSO | Sigma-Aldrich Cat# 276855 | Sigma-Aldrich Cat# 276855 | 0.05% |
| Chemical compound, drug | PD-0325901; MEKi; MEK inhibitor | Cayman Chemical Cat# 13034 | Cayman Chemical Cat# 13034; CAS: 391210-10-9 | 1.5 nM |
| Sequence-based reagent | HD3 dsRNA library; dsRNAs for combinatorial library | other; PMID: 26912791 | | in house synthesized dsRNA library |
| Software, algorithm | R | https://cran.r-project.org/; R Project for Statistical Computing | nif-0000–10474; OMICS_01147; RRID: SCR_001905 | |
| Software, algorithm | custom R code | this paper; https://github.com/boutroslab/Supplemental-Material/tree/master/Heigwer_2018 | | |
| Software, algorithm | Perl | http://www.cpan.org; Comprehensive Perl Archive Network | nif-0000–30267; RRID: SCR_007253 | |
| Other | source data; well-wise feature data | this paper; https://doi.org/10.6084/m9.figshare.6819557 | | |
| Other | Flybase gene annotation database | http://flybase.org/; PMID: 30364959 | RRID:SCR_006549 | |

## Cell line

All cells used in this project were from the same culture of the, serum-free medium adapted, *Drosophila melanogaster* S2 cell line (S2) and will be referred to as S2 cells (Schneider's *Drosophila* Line 2 [D. Mel. (2), SL2] (ATCC CRL1963) from ThermoFisher (Waltham, MA, *Billmann et al., 2018*; *Horn et al., 2011*; *Fischer et al., 2015*).

## Genome-wide RNAi library

We used a genome-wide *D. melanogaster* dsRNA library (HD3-dsRNA library) in this study, as previously described (*Billmann and Boutros, 2017*; *Horn et al., 2011*). The library contains 28941 dsRNA reagents targeting 14242 unique gene IDs in the *D. melanogaster* genome and contains two sequence independent reagents targeting 13617 IDs twice and the remaining genes once. The reagents were optimized for the BDGP5 mRNA annotations in *D. melanogaster* by for example avoiding CAN repeats and non-unique sequences (off-targets). 250 ng dsRNA, synthesized as described previously, were aliquoted to 384 Greiner µClear plates prior to the image-based assay at a mass of 250 ng/well. A table containing all sequences that were used in the genome-wide RNAi screen can be found in *Supplementary file 4*. Another table containing sequence IDs (HD3) that were used in the combinatorial RNAi screen can be found in *Supplementary file 5*.

## Image-based RNAi screening

dsRNA reagents dissolved in water were spotted into barcoded 384-well microscopy plates (Greiner µClear, black, flat-transparent-bottom, Ref: 781092, Greiner Bio One International GmbH, Frickenhausen, Germany) to reach a final mass of 250 ng dsRNA per well (5 µl of a 50 ng/µl solution). Express V medium (Gibco, Ref: 10486–025, Life Technologies GmbH, Darmstadt, Germany) with 10% Glutamax (Gibco, Ref: 35050–061) was pre-warmed to 25°C and 30 µl were dispensed on top of the spotted dsRNA using a MultiDrop Combi dispenser and standard cassette (Thermo Fisher Scientific, Ref: 5840400, Life Technologies GmbH, Darmstadt, Germany).

10 µl of pre-diluted S2 cell suspension were seeded to a final concentration of 9000 cells/well into the prepared assay plates using MultiDrop Combi dispensing under constant stirring of the suspension in a sterile spinner flask (Corning, Ref: CLS4500500, Kaiserslautern, Germany). After cell addition, the assay plates were heat sealed using a PlateLoc (peelable seal, 2.3 s at 180°C, Agilent Technologies Deutschland GmbH and Co. KG, Waldbronn, Germany) and centrifuged at 140x g for 60 s. Cells were incubated for 24 hr at 25°C without $CO_2$ adjustment.

After 24 hr incubation, plates with growing cells were opened and small molecule treatment was performed. The concentration of applied compound is outlined with the separate experiments in the following paragraphs. Per well 5 µl of a solution containing 5% DMSO (Sigma Aldrich, Ref: 41644–1 l, Merck KGaA, Darmstadt, Germany) in medium, or the MEK-inhibitor PD-0325901 (Cayman chemical, Ref: CAY-13034–5, Biomol GmbH, Germany) dissolved in 5% DMSO in medium, were added to achieve a final assay concentration of 0.5% DMSO and varying small molecule concentrations. After compound addition, plates were sealed again and incubated at 25°C without $CO_2$ adjustment for 48 hr, 72 hr or 96 hr depending on the experiment.

Assays were stopped after the second incubation period by fixation using a robotics procedure on a CyBioWell vario (384-well pipetting head, Analytic Jena AG, Jena, Germany). The supernatant was removed, and cells were washed with 50 µl PBS (Sigma Aldrich, Ref: P3813-10PAK) per well. After addition of 40 µl Fix-Perm solution (4% Para-formaldehyde (Roth, Ref: 0335.3, Karlsruhe, Germany); 0,3% Triton X-100 (Sigma Aldrich, Ref: T8787-250ml); 0,1% Tween20 (Sigma Aldrich, Ref: P1379-100ML); 1% BSA (GERBU Biotechnik GmbH, Ref: 1507.0100, Heidelberg, Germany)), plates were incubated for 60 min at RT and then washed twice with 50 µl of PBS. 50 µl of PBS were added again and plates were stored at 4°C before staining. For staining, remaining PBS was removed and fixed cells were first blocked by adding 30 µl of blocking solution (4% BSA; 0,1% Triton X-100, 0, 1% Tween20) and incubated for 30 min at RT. Next, the blocking buffer was removed and 10 µl of staining solution (1:4000 Hoechst (Thermo Scientific, Ref: H1399, Life Technologies GmbH, Darmstadt, Germany), 1:1500 primary FITC labelled anti α-tubulin antibody (Sigma Aldrich, P1951), 1:6000 Phalloidin-TRITC conjugate (Sigma Aldrich, F2168-.5ml) in 1x blocking buffer) were added. After addition of the staining solution plates were incubated for 60 min at RT in the dark. After staining, 30 µl of PBS were added and the staining solution was removed. After two additional washing steps with 50 µl PBS another 50 µl of fresh PBS were added per well and stored at 4°C until imaging.

## Genome-wide drug-gene interaction screening

We performed genome-wide RNAi screens in combination with drug and solvent control treatment to verify dsRNA reagent efficiency, identify candidate genes for combinatorial screening and to find which genes react most differentially to the *Dsor1* inhibitor (PD-0325901) treatment. Four sets of 88

× 384 well Greiner µClear plates were spotted with the HD3 library, 5 µl of 50 ng/µl dsRNA in each well. The HD3 library is arrayed to target one gene with one dsRNA design per well. Two additional plates, containing only controls were added to control assay reproducibility, robustness and effect size. Controls were chosen to spread over the complete dynamic range of cell fitness. dsRNAs against *RLUC* and *GFP* expressing plasmids serve as non-targeting negative controls, such that we could control for unspecific dsRNA induced phenotypes. dsRNA containing plates were thawed, seeded with cells and left for 24 hr at 25°C without $CO_2$ adjustment for incubation prior to drug treatment. Plates were opened and 5 µl of 15 nM PD-0325901 in 5% DMSO were added resulting in a final assay concentration of 1.5 nM PD-0325901 in 0.5% DMSO in medium. Cells were left to incubate for another 72 hr at 25°C without $CO_2$ adjustment prior to fixation, staining and imaging. Images were acquired using the standard protocol described below with low illumination timings (DAPI: 100 ms, Cy3: 200 ms, FITC: 300 ms). The resulting images were analyzed in line with the acquisition using the standard image analysis pipeline and progress was monitored using our automated analysis pipeline as described below.

## Combinatorial RNAi screening under differential time and treatment conditions

The design of the library for combinatorial screening is described in a separate paragraph. 168 genes were chosen for design of a combinatorial RNAi library. The dsRNA sequences that were used in the combinatorial library can be found in *Supplementary file 4* and *5*. All used labware and reagents, which are not further detailed here have been the same as in previous experiments. The library contained 12 batches for screening, each comprising 80 × 384 well Greiner µClear plates spotted with 250 ng dsRNA/well dissolved in 5 µl of DNase, RNase-free water. dsRNAs were obtained from the HD3-library templates and synthesized accordingly. To avoid contaminations, all dsRNAs were sterile filtered using Steriflips-0.22µm (Merck Millipore, Ref: SCGP00525, Darmstadt, Germany) for the query dsRNAs and MultiScreenHTS-GV 0.22 µm filter plates (Merck Millipore, Ref: MSGVN2250) for the target dsRNAs. Genes were divided into target and query genes based on prior knowledge on key pathway components and screened a matrix of 76 genes combined with 168 genes. All query genes were included in the list of target genes. We screened each target gene in two sequence independent designs and each query gene in one design. This way, we screened 25536 dsRNA combinations (12768 gene pairs) in each batch. Combinatorial dsRNA spotting was achieved with combining the query and target master plates such that 2.5 µl of each query dsRNA were spotted onto 2.5 µl target dsRNA using a Beckman FX robotic liquid handling station (Beckman Coulter, Krefeld, Germany). In order to control for RNAi induced phenotypes and per-plate batch effects control dsRNAs against *Dsor1*, *drk*, *Diap1*, *RasGAP1*, *Pten*, *pnt*, *Pvr*, *Rho1* and *RLUC* expressing plasmid were spotted on each plate and not paired with a second query dsRNA perturbation. Two control plates containing only the target gene dsRNA reagents with 250 ng dsRNA per well complete one screening batch of 80 plates and controlled for screening batch effects. To achieve differential treatment and time resolution, 12 screening batches were prepared. They were divided into two groups of six batches, which then were treated under different conditions in duplicate. six library batches are needed to screen two conditions (1.5 nM PD-0325901% and 0.5% DMSO) at three time points (fixation 48 hr, 72 hr, 96 hr after small molecule addition), all in all comprising 480 screened plates. The entire experiment was repeated twice. This way we screened 960 × 384 well plates.

The assay workflow followed the same procedures as outlined above. Briefly, 9000 cells per well were seeded onto 384-well Greiner µClear plates for microscopy, which were pre-spotted with a combinatorial dsRNA library. After centrifugation, plates were sealed and left to incubate for 24 hr at 25°C prior to compound addition. Therefore, a PD-0325901 dilution (15 nM in medium with 5% DMSO), and a 5% DMSO-only dilution in medium were prepared and added to the opened plates. This resulted in either 1.5 nM PD-0325901 or 0.5% DMSO in-assay concentrations. Plates were sealed again using a heat sealer and left to incubate until the experiment was stopped by fixation after 48 hr, 72 hr and 96 hr, respectively. Stained plates were imaged using an InCell Analyzer 2200 automated fluorescence microscope according to the protocol described above with 20x magnification, three channels per field and four fields per well. Resulting images were analyzed using the R/EBImage based pipeline described below.

## Imaging

All plates were imaged using the same protocol. There, an InCell-Analyzer 2200 automated fluorescence microscope (GE Healthcare GmbH, Solingen, Germany) with a Nikon SAC 20x objective (NA = 0.45) was used. The microscope was adjusted to scan Greiner µClear plates by setting the bottom height to 2850 µm and the bottom thickness to 200 µm and the laser autofocus function was applied to identify the well bottoms with attached cells. This Z-position was used for image acquisition in three fluorescence channels: Hoechst (excitation: $390 \pm 18$ nm, emission: $435 \pm 48$ nm) at 400 ms exposure (100 ms in dose response experiments and genome wide screens), Cy3 (excitation: $475 \pm 28$ nm, emission: $511 \pm 2$ nm 3) at 300 ms exposure (200 ms in dose response experiments and genome wide screens) and FITC (excitation: $542 \pm 27$ nm, emission: $597 \pm 45$ nm) at 300 ms exposure (300 ms in dose response experiments and genome wide screens). Four fields of view were imaged per well at 20x magnification each representing a 665.60 µm x 665.60 µm area covered (approximately 20% of total well area) by $2048 \times 2048$ pixels. For plate handling, the microscope was equipped with a KinedX robotic arm (PAA Scara, Peak Analysis and Automation Ltd, Hampshire, UK) allowing a fully automated image acquisition.

## Automated image processing and high-throughput image analysis

Plates were imaged and analyzed in batches of 40 plates and a custom pipeline allowed parallel image processing and analysis by bundling images of fields and channels of several wells. During imaging an automated pipeline scheduled the processing of image files for each field of view through the following analysis workflow, here described representatively for one field. Raw images of three channels with a size of 8.4 MB (16-Bit grey scale, $2048 \times 2048$ pixels) per image were captured with the InCell Analyzer 2200 software and saved as TIFF files on a server cluster for image processing and analysis. The image processing- and analysis pipeline covered two main blocks, first a sequence of pre-processing steps which was followed by extraction of phenotypic features from single cells. First, the images were read in and each channel was assigned to the subcellular structure that was selectively stained with the above described assay (Hoechst: DNA, Phalloidin-Cy3: F-actin, α-tubulin-FITC: tubulin). To identify cell and nuclei boundaries, a duplicate image of each channel is ln transformed, scaled between 0 to 1 and smoothened by a Gaussian filter using a sigma of one. This reduced optical noise, improved the dynamic range and smoothened the image gradients for further segmentation by thresholding. For segmentation, the normalized actin and tubulin images were binarized by global thresholding. Second, the cell nuclei were identified by applying a local adaptive average threshold to binarize the DNA channel nuclei image and assigning objects. The resulting binary image was then subjected to morphological operations of opening and hull filling such that filled objects with smoothly roundish outlines result. Offsets for segmentation were varied if the channels surpassed certain thresholds. If more than 30 nuclei were counted per field, the objects were subjected to further propagation of nuclei objects into the an á priori defined cell body mask. Starting from the nucleus objects as seed regions, the cell bodies are segmented by propagating the nuclei objects into foreground area (*Carpenter et al., 2006*). This strategy allowed to identify single cells and corresponding nuclei as objects. Using the segmented object outlines as masks, features on each object and channel were calculated on the original image using the R/Bioconductor package EBImage (*Pau et al., 2010*).

Specifically, numeric descriptors for five feature classes are defined in the computeFeatures function from EBImage (*Supplementary fle 6*): (i) shape features which quantify the shape of cells and nuclei, (ii) basic features that describe the summary statistics, such as 5% quantiles, of pixel intensity within the borders of the object, (iii) moment features that describe the spatial orientation of the objects, (iv) Haralick features derived from a pixel intensity co-occurrence matrix as texture descriptors (*Haralick et al., 1973*) and (v) social features such as distance to the first 20 nearest neighboring cells. Social features are derived by a k-nearest-neighbor search based on the geometric center points of single cells. Single-cell data were stored and aggregated to well averaged data by calculating the trimmed mean (q = 0.01) of all cells belonging to all fields of one well and its standard deviation.

## Data processing and normalization

For the analysis of the genome-wide drug-gene interaction screens the following analysis strategy was pursued: feature data was collected in a data frame containing per-well aggregated values as trimmed mean and standard deviation. This data frame was then reformatted to a 4-dimensional data cube featuring the dimensions: feature, plate, well, screen. Per feature, the feature's minimum value was added to each value prior to $\log_n(x + 1)$ transformation to approach the features' histogram to normal distribution. Following transformation, each plate in each screen was normalized separately for each feature by B-Score normalization (*Ljosa et al., 2013*; *Mpindi et al., 2015*). The B-Score normalization centers and scales the data to be the residuals of the median polish divided by the median absolute deviation (mad) across all values of the plate and thus be symmetrically centered around zero and scaled in units of the mad. Here, 38 representative features were chosen based on their biological significance (our ability to refer them back to cellular phenotypes) and their biological reproducibility between the two mock (DMSO) treated replicate screens and their information content, as measured by added variance (*Supplementary file 6*).

For the combinatorial screens, the obtained data frame containing rows for each well and columns for each feature. In addition, we added well, plate and batch identifiers as annotation columns. Data acquired from single cells was aggregated by calculating the trimmed mean (q = 0.01) for each feature extracted in the respective well together with its standard deviation. This way, outliers, produced by over or under segmentation of cells, were mostly excluded from further analysis. Data was normalized by dividing each feature in each plate by the median of the non-targeting control wells (if that was not zero). Further, the values of each feature were transformed on a logarithmic scale using the generalized logarithm with c being the 3% quantile of the features value distribution over all values (*Caicedo et al., 2017*; *Fischer et al., 2015*). For each feature, data was subsequently scaled and centered around 0 by using the robust Z-transformation, where the feature median is subtracted from each value and the result was divided by the median absolute deviation ($x'=(x-median(x))/mad(x)$). After that, all features were in normalized units of median absolute deviation from the median of that feature and normalized per plate. The normalized feature vector provided the basis to all further analyses.

## Candidate selection for combinatorial RNAi screening

The metrics used for judging the quality of dsRNA reagents and to assess the gene's suitability for the combinatorial screen are summarized in *Supplementary file 1*. For this purpose, several metrics have been deployed. Summarized, the applied metrics were used to assesses for each individual gene in the genome-wide HD3 library (i) the quality of the RNAi reagent, (ii) the effect size of the induced phenotypes under solvent control treatment as well as the differential effect size of the treatment-sensitive phenotype between small molecule treatment and control conditions, (iii) the quality of the target gene as a candidate for gene-gene-drug combinatorial screening. Effect size was quantified using the Euclidean distance ($\sqrt{\sum_{i=1}^{n}(x_i - y_i)^2}$) between sample and control measurements under different conditions. Quality of dsRNAs was assessed by calculating Pearson correlation coefficients between phenotypic profiles of biological and dsRNA design replicates. The quality of genes as screening candidates was assessed by gene expression analysis and literature analysis. The Q1 metric shows the strength of a knockdown induced profile when compared to the non-targeting control knockdown (here: GFP). This was calculated as the Z-Score normalized Euclidean distance of the genes profile to the control profile and can be used to inform if a phenotype of a gene is exceptionally strong or weak. In general, strong phenotypes (Q2) were preferred since they were more robust to experimental noise and are likely to engage in many genetic interactions (*Costanzo et al., 2010*). Q3 gives to what extent the phenotypic profile of those genes' knockdown changes upon drug treatment. An ideal candidate for drug gene interaction screening shows a high value in this metric. Q4 and Q5 allow inferring the reproducibility of the measured phenotype by comparing the correlation of two sequence-independent dsRNA designs targeting one gene and the correlation of one design across screen replicates, respectively. There, 7957 genes were targeted by designs whose feature vectors correlate with PCC >0.5 while 17263 designs were reproducible between screens (PCC >0.5). Q6 was used to infer if the respective gene is expressed under the screened conditions (S2 cells, 4 days in culture in Express-V medium). 12567 genes (88% of all genes screened) had a $\log_2$ normalized read count greater than 0. In contrast, the knowledge sum in Q7 was used to

avoid over enrichment of well-characterized genes in the final combinatorial library. The 'unknown' was defined by means of assigning each gene a score describing how well it has been studied and characterized. Therefor the Gene Ontology terms associated to each *Drosophila* gene were downloaded from Flybase. In Flybase, each ontology term is annotated with evidence codes as provided by the gene ontology consortium (*Ashburner et al., 2000*). Each of these codes was then used to assign weights to the ontology terms for each gene (*Supplementary file 7*). Ontology terms derived from experimental evidence, such as genetic interactions, direct assays or physical interactions were assigned the highest weight while computational annotations were weighted the lowest. For each gene, the sum of ontology terms was computed and used as a proxy for the current state of its functional characterization. For example, the cell fate determining receptor *Notch* is the most well studied gene with a score of 973, while all genes have an average score of 34.7 and the third quartile ends at 41. This means that only a minor fraction of genes is as well studied as *Notch* and most genes can be accounted as uncharacterized if their score is beneath 100 (90 % quantile). An example for such a gene is *tzn* with a knowledge sum of 14. Only known fact about *tzn* is its function as Hydroxyacylglutathione hydrolase in response to hypoxia (*Neely et al., 2010*; *Jha et al., 2016*). For screening, genes with a low knowledge sum were preferentially chosen.

## Modeling of genetic interactions

The data frame with normalized feature data per well was reformatted into a five-dimensional data cube representing the experimental design. The dimensions are: target dsRNA (two entries for each gene), query gene, time, treatment and feature. The data cube was further subjected to genetic interaction analysis following the protocol established by Bernd Fischer (*Fischer et al., 2015*; *Horn et al., 2011*; *Laufer et al., 2013*). There, genetic interactions are defined as the residuals of a modified median polish over the double perturbation matrix of one replicate, feature, treatment and time point. The median polish presents a robust linear fit ($M_{ij} = m_i + n_j + \pi_{ij} + \varepsilon$) that lifts the main effects (m, n) of each query such that it resembles the value of a single gene knockdown. The residuals of this fit scaled by their median absolute deviation are defined as π-scores. π-scores further provide us with a quantitative measure of genetic interaction following the multiplicative model plus some error term (ε) estimating the experimental noise. There, the interaction of two genes is defined as the deviation of the measured combined phenotype ($M_{ij}$) from the expected phenotype for a target-query gene pair *i* and *j*. The expected phenotype is defined as the product of the two independent single knockdown phenotypes. The resulting π-scores are then collected for all replicates (dsRNA and experimental, each interaction is measured four times). The significance of their mean over all measured scores is estimated by a moderated students t-test as is implemented in the R-package limma. There, the t-test is adapted for situations where a small amount of observations is tested in many tests, normally causing large test variability, using an empirical Bayes variance estimator. p-values were adjusted using the methods of Benjamini Hochberg (*Benjamini and Hochberg, 1995*). From there on, adjusted p-values can be treated as false discovery rates. The FDR estimates the chance that the finding was observed by random chance given the entire dataset. This described procedure was applied to quantitatively calculate genetic interactions for each phenotypic feature.

## Modeling of treatment-sensitive interactions

To identify a hit-list of condition-sensitive gene-gene interactions, we tested whether the changes of genetic interactions over time and between different conditions could be quantitatively described by a multi factorial linear model. This would provide the possibility to (i) quantify the time dependence of an interaction and (ii) to measure the phenotypic difference between treatment conditions with high confidence. For every gene-gene combination [i, j] screened across time and chemical treatment, we used a two-factor robust linear model, which we termed model of differential interactions (MODIFI), to estimate the predictive strength and influence of time and differential compound treatment on the π-score ($\pi_{ij} = c_{ij} + \sigma_{ij} * time + \delta_{ij} * treatment + \varepsilon_{ij}$). Therein, the coefficient $\sigma_{ij}$ models the time dependence, $\delta_{ij}$ models the quantitative offset between treatments, c estimates the intercept and the residual $\varepsilon_{ij}$ estimates the error of fit for each combination of the target gene i and the query gene j. σ and δ are thus parameter estimates that uniquely describe the behavior of each gene-gene interaction. A separate model was fitted for every feature and every gene-gene combination using the "rlm" function of the R/MASS package. A p-value denoting the predictive power of

each covariate (time, treatment) on the π-score was estimated by a robust F-test as implemented in the function f.robftest function from the R/sfsmisc package. For statistical assessment the difference in interaction strength is used, as opposed to the interaction in a single condition. This way MODIFI identifies a great number of treatment sensitive interactions where the interaction score in each isolated condition is small, but the difference between conditions is significant. This resulted in a data frame that, for each gene-gene combination and each feature, contains a p-value for each covariate and its estimate. The p-value was multiple testing corrected by FDR analysis (Benjamini and Hochberg, 1995). Interactions with an FDR<0.1 in either term (time, treatment or both) were called significant. The FDR threshold also served as the basis for classifying context-dependent interactions into the different classes (*Figure 1C*). Interactions are time dependent if the adjusted p-value for the time term is below 0.1, treatment sensitive when the adjusted p-value for the treatment term is below 0.1, and context-independent else. Interactions are aggravating when the π-score is negative and alleviating if it is positive.

## qPCR analysis

Quantitative real-time PCR (qPCR) was used to analyze the transcriptional response following Rel/pnt co-RNAi. To this end, as $5*10^5$ cells / well were seeded in 630 µl ExpressFive (Gibco) culture medium and reverse transfected with 14 µg dsRNA. All dsRNAs denoted with #2 were used in three biological replicates and combinatorial RNAi was achieved by mixing 7 µg of dsRNA targeting each gene (*Supplementary file 8*). After 72 hr incubation (25°C, no CO2 adjustment), cells were washed once in 750 µl PBS (Gibco) and lysed in 350 µl RLT buffer shipped with the RNAeasy-mini Kit (Qiagen). RNA was then purified from all samples according to manufacturer's standard instructions for spin column purification. An optional DNase digestion step was performed using the RNase-Free DNase Set (Qiagen). Samples were prepared for qPCR by reverse transcription of 1 µg of RNA using RevertAid H minus First strand cDNA Synthesis kit (Thermo scientific) according to the manufacturer's standard protocol. A qPCR reaction was prepared using PrimaQuant 2x qPCR-Mastermix (Steinbrenner) by mixing 5 µl of sample (1:10 diluted cDNA) with 5 µl of Mastermix (including 0,3 µM of forward and reverse primer, *Supplementary file 9*) on a 384-well qPCR plate (LightCycler 480 Multiwell Plate 384, white, Roche). The plate was then centrifuged (2 min, 2000 rpm) and processed for qPCR in a Roche 480 LightCycler using the following PCR program: (i) 10 min at 95°C, (ii) 15 s at 95°C, (iii) 60 s at 60°C, repeat step ii) and iii) 40 times and measure fluorescence at 494 nm-521 nm during step iii). Melting curve analysis of each sample was performed to assess reaction quality. Relative expression of each gene in each sample (normalized to rps7 expression) was analysis as $log_2$-foldchange over RLUC dsRNA-treated samples (*Nolan et al., 2006*; *Schmittgen and Livak, 2008*). qPCR primers were designed using the GETprime web service (*Gubelmann et al., 2011*).

For analysis, all genes in the combinatorial library were annotated manually using FlyBase and literature annotations (*Marygold et al., 2013*).

A more detailed description of all methods including those for supplementary materials can be found in Appendix 1.

All code used for the analysis presented in this study is available for download at: https://github.com/boutroslab/Supplemental-Material/tree/master/Heigwer_2018 (*Heigwer, 2018*; also forked at https://github.com/elifesciences-publications/Supplemental-Material/tree/master/Heigwer_2018).

All raw data is available at: https://doi.org/10.6084/m9.figshare.6819557

## Acknowledgements

We thank Benedikt Rauscher, Maja Funk, Niklas Rindtorff, Katharina Imkeller, Fillip Port, Jan Gleixner, Josephine Bageritz, Bojana Pavlovic for critical comments on the manuscript. We thank Luisa Henkel, Tianzuo Zhan and the members of the Boutros laboratory for fruitful discussions and valuable inputs into the study. FH was supported by family support and a travel grant provided by HBIGS. MB is an investigator of CellNetworks. This work was supported by the ERC Advanced Grant 'SYNGENE'.

## Additional information

### Funding

| Funder | Grant reference number | Author |
| --- | --- | --- |
| H2020 European Research Council | SYNGENE | Florian Heigwer<br>Christian Scheeder<br>Thilo Miersch<br>Barbara Schmitt<br>Claudia Blass<br>Mischan Vali Pour Jamnani<br>Michael Boutros |

The funders had no role in study design, data collection and interpretation, or the decision to submit the work for publication.

### Author contributions

Florian Heigwer, Conceptualization, Data curation, Software, Validation, Investigation, Visualization, Methodology, Writing—original draft, Writing—review and editing; Christian Scheeder, Investigation, Visualization, Writing—original draft, Writing—review and editing; Thilo Miersch, Barbara Schmitt, Claudia Blass, Investigation, Writing—review and editing; Mischan Vali Pour Jamnani, Validation; Michael Boutros, Conceptualization, Supervision, Methodology, Writing—original draft, Project administration, Writing—review and editing

### Author ORCIDs

Florian Heigwer http://orcid.org/0000-0002-8230-1485
Christian Scheeder http://orcid.org/0000-0003-1786-4916
Michael Boutros http://orcid.org/0000-0002-9458-817X

### Decision letter and Author response
Decision letter https://doi.org/10.7554/eLife.40174.045
Author response https://doi.org/10.7554/eLife.40174.046

## Additional files

### Supplementary files
• Supplementary file 1. Metrics that were used to assess candidate genes from genome wide gene-drug interaction screens
DOI: https://doi.org/10.7554/eLife.40174.031

• Supplementary file 2. Manually curated list of gene annotations used for all analyses
DOI: https://doi.org/10.7554/eLife.40174.032

• Supplementary file 3. Detailed description of phenotypic features used within the combinatorial RNAi screen
DOI: https://doi.org/10.7554/eLife.40174.033

• Supplementary file 4. Genome wide dsRNA library annotation
DOI: https://doi.org/10.7554/eLife.40174.034

• Supplementary file 5. Annotation file for the combinatorial dsRNA library
DOI: https://doi.org/10.7554/eLife.40174.035

• Supplementary file 6. Detailed description of phenotypic features used within the genome-wide RNAi screens
DOI: https://doi.org/10.7554/eLife.40174.036

• Supplementary file 7. Weights to GO-term confidence levels
DOI: https://doi.org/10.7554/eLife.40174.037

• Supplementary file 8. List of dsRNAs used for all follow-up experiments
DOI: https://doi.org/10.7554/eLife.40174.038

• Supplementary file 9. List of qPCR primers used for all follow-up experiments

DOI: https://doi.org/10.7554/eLife.40174.039

• Transparent reporting form

DOI: https://doi.org/10.7554/eLife.40174.040

### Data availability

MODIFI data has been uploaded to figshare (https://doi.org/10.6084/m9.figshare.6819557). A code package (Florian Heigwer, 2018) is available via GitHub (https://github.com/boutroslab/Supplemental-Material/tree/master/Heigwer_2018; copy archived at https://github.com/elifesciences-publications/Supplemental-Material/tree/master/Heigwer_2018).

The following dataset was generated:

| Author(s) | Year | Dataset title | Dataset URL | Database and Identifier |
|---|---|---|---|---|
| Heigwer F, Scheeder C, Miersch T, Schmitt B, Blass C, Pour-Jamnani MV, Boutros M | 2018 | MODIFI data: from Time-resolved mapping of genetic interactions to model rewiring of signaling pathways | https://doi.org/10.6084/m9.figshare.6819557 | figshare, 10.6084/m9.figshare.6819557 |

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

## Appendix 1

DOI: https://doi.org/10.7554/eLife.40174.041

## Detailed methods

### Testing the sensitivity of S2 cells to the MEK inhibitor PD-0325901 (PD)

Prior to large scale RNAi screening and RNA-Seq the sensitivity of S2 cells towards small molecule MEK inhibitor PD-0325901 (Cayman chemical, Ref: CAY-13034–5, Biomol GmbH, Germany) was tested under three different conditions: (i) normal treatment regimen, (ii) recovery after drug wash-out, (iii) treatment with used drug medium. All cells used in this project were from the same culture of the, serum free medium adapted, Drosophila *melanogaster* S2 cell line (S2) and will be referred to as cells (Schneider's Drosophila Line 2 [D. Mel. (2), SL2] (ATCC CRL1963) from ThermoFisher (Waltham, MA).

### Experimental setup and drug treatment regimen

Cells were seeded in 2 Greiner µClear plates (Ref: 781073, Greiner Bio One International GmbH, Frickenhausen, Germany), pre-spotted with 250 ng/well anti-GFP dsRNA and left to incubate for 24 hr at 25°C. After incubation, the cells were treated with a gradient of PD-0325901 in medium (Express V, serum free cell culture medium +10% Glutamax, Gibco, Ref: 10486–025 and Ref: 35050–061, Life Technologies GmbH, Darmstadt, Germany) using robotic liquid handling. The gradient was pipetted as a 2-fold serial dilution in 100% DMSO from a 10 mM stock of PD-0325901 using a Beckmann NX-P pipetting robot (Beckman Coulter, Brea, California, USA). The prepared gradient was then diluted in 384 well format 1:20 in medium reaching a pre-assay concentration of 5% DMSO in medium. Addition of 5 µl gradient dilution mix to the cell culture adds another 1:10 dilution step yielding an in-assay concentration gradient ranging from 50 µM to 0.024 nM PD-0325901 in 0.5% DMSO in medium. Thus, each column was treated with one concentration of PD-0325901. Column 1 and 24 were left mock treated with 0.5% DMSO. Plates were sealed again and incubated at 25°C for 96 hr. After completed incubation one plate was subjected to fixation and staining while the medium of the other plate was carefully removed and dispensed on a plate containing cells freshly seeded 24 hr in advance. 50 µl of fresh medium was then added to the emptied plate. Both, the plate with fresh medium and the plate with fresh cells and used medium were then incubated for additional 72 hr prior to fixation and staining. This results in three dose response analyses. (1) with normal assay protocol, (2) where the cells could recover from the treatment to test if the phenotypes are temporal and (3) where the remaining activity of PD-0325901 was tested after 96 hr of incubation on cells. All experiments in this setup have been replicated twice.

### Data normalization

All plates were subjected to standard fixation, staining and imaging protocols and analyzed using automated image analysis as described in the main methods. Data were triaged by removing NA value containing wells and single cell data was aggregated by calculating the mean per field of view. The data were scaled per feature on a scale from 0 to 1 and subjected to dose-response modelling.

### Dose-Response modeling and statistical analysis

Dose-response-modelling was carried out using the drm function for dose response models from R package drc. For modelling, the replicates and fields per well were used as independent measurements to inform the model. Dose response data was fitted to a four-parameter log-logistic function estimating the parameters slope, lower limit, upper limit and $ED_{50}$. We used this model to accurately estimate the $ED_{50}$ concentration of cell growth inhibition. There the model provided us with a fixed value plus and minus a confidence interval in which the coefficient was estimated. The resulting model was assessed via plotting the dose

response curves for different features, including all measured points with their mean and s. e.m. and the $ED_{50}$ was further used in all subsequent experiments.

## Western Blot analysis of in vitro PD-0325901 activity

Inhibition of Dsor1 kinase activity by PD-0325901 was tested by assaying levels of phosphorylated rl. Therefore, S2 cells were seeded as $4.5 \times 10^5$ cells per well in a 24-well cell Culture plate (Greiner, Ref: 2511, Greiner Bio One International GmbH, Frickenhausen, Germany) in 700 µl Express Five cell culture medium (Gibco, Ref: 10486–025, Life Technologies GmbH, Darmstadt, Germany) supplemented with 10% Glutamax (Gibco) in four identical wells. Cells were incubated at 25°C for 24 hr prior to compound addition (150 nM PD-0325901, 0.5% in assay) in well A. After another 24 hr, well B was treated. This procedure was repeated for well C and D. After additional 24 hr of incubation, cells in all wells were lysed and western-blot analysis was carried out as follows. After washing the cells once with 500 µl PBS (Life Technologies, Ref: 10010015, Gibco), 50 µl of ice-cold RIPA-lysis buffer (50 mM Tris HCl, pH 8.0, 150 mM NaCl, 1 % NP-40, 2 mM EDTA,1 x Protease Inhibitor, 2% Phosphatase inhibitor cocktail II and III, Sigma, Merck KGaA, Darmstadt, Germany) were added and cells left to lyse for 30 s on ice. The lysate was collected in fresh 1.5 ml tubes (Eppendorf, Eppendorf AG, Hamburg, Germany) and samples were centrifuged (21000 rpm, 5 min, 4°C). 20 µl of supernatant were mixed with 5 µl 5x Laemmli buffer and incubated 10 min at 95°C. Another 2 µl of each sample were used to measure protein content after manufacturer's instructions for BCA assays 96-well plates (Pierce, Ref: 10741395, Thermo Scientific, Ref: H1399, Life Technologies GmbH, Darmstadt, Germany). Samples were then loaded on a 12-well NuPAGE 4–12% Bis-Tris Gel (Novex, Ref: NP0323BOX, Life Technologies GmbH, Darmstadt, Germany) and left for electrophoresis for 30 min at 80 V constant voltage followed by 60 min at 120 V constant voltage in $1 \times$ 3-(N-morpholino) propanesulfonic acid (MOPS) buffer (40 mM MOPS, 10 mM NaAc, 1 mM EDTA). Following electrophoresis, proteins were transferred onto a methanol activated PVDF-membrane (Immobilon-P, Millipore, Ref: IPVH00010, Merck KGaA, Darmstadt, Germany) by tank-blotting in 1 x Transfer buffer (25 mM Tris base, 192 mM glycine, 10% methanol) at 35 V for 90 min. After transfer, the membrane was shortly washed in 1 x TBST (137 mM NaCl, 2.7 mM KCl, 19 mM Tris Base) and incubated for 60 min with constant shaking at room temperature (RT) in 5% skimmed milk (Sigma, Ref: 70166–500G, Merck KGaA, Darmstadt, Germany). After washing the membrane 5 times 5 min at RT with 1 X TBST, the membrane was incubated overnight in a 1:2000 dilution of anti pp-p44/42 rabbit monoclonal antibody (Cell-Signaling, Ref: 4370, Cell Signaling Technology, Frankfurt, Germany) in 5% BSA (Sigma, Ref: A9085-25G). Following washing the membrane five times, 5 min in 1 x TBST it was incubated 60 min in 1:10000 anti-rabbit IgG-HRP conjugate (from Donkey) (Amersham ECL igG-HRP Conjugate, GE Healthcare). After an additional washing step, the membrane was developed on Hyperfilm ECL (Sigma Aldrich, Ref: GE28-9068-36) using Immobilon ECL substrate (Merck Millipore, Ref: WBKLS0100). The membrane was then stripped in 1 x Re-Blot Plus strong (Merck Millipore, Ref: 2504) for 15 min, RT followed by washing once 5 min in 1 x TBST. The membrane was again blocked 60 min in 5% skimmed milk in TBST, stained with 1:1000 anti p44/42 rabbit monoclonal antibody (Cell-Signaling, Ref: 4695), developed the same way as with the first antibody and stripped again. The same procedure was repeated using a 1:2000 dilution of anti α-tubulin monoclonal rabbit antibody (Cell-Signaling Ref: 2144).

## Transcriptional Profiling of S2 cells

RNAi most effectively induces a visible phenotype when targeting highly expressed genes. Thus, prior to image based screening and to gain information about ongoing cellular processes, differential expression analysis was performed under different treatment conditions using conventional bulk RNA-seq analysis.

## dsRNA preparation

$10 \times$ 96-well plates (cell culture grade, flat bottom, transparent, Greiner, Ref: 655083) were prepared by dispensing 18 µl of a 50 ng/µl solution (equaling ~900 ng dsRNA per well) of anti-

RLUC dsRNA in culture medium (Express V, serum free cell culture medium +10% Glutamax, both Gibco) into each well using a multichannel manual pipette. Additional 95 µl pre-warmed (25°C) medium were added to each plate using a MultiDrop dispenser (Thermo Scientific, low speed, 96-well standard plate, full plate, 20 µl pre-dispense and 16 mm offset). All further MultiDrop dispensing steps have been carried out using the same settings if not denoted different. The same cell culture medium was used in all the following experiments and is referred to as medium. All cells used in this project were from the same culture of the, serum free medium adapted, *Drosophila melanogaster* S2 cell line (S2) and will be referred to as cells.

## Cell culture

S2 cells (at passage 21, counted via a Nexcelom Cellometer Auto 1000 for 90% viability, Ref: CETHT4SD100002, Cenibra GmbH) were seeded in all plates at 40000 cells/well in 35 µl medium using a MultiDrop dispenser. Plates were sealed using a heat plate sealer (PlateLoc, Agilent Technologies Deutschland GmbH and Co. KG, Waldbronn, Germany) and centrifuged at 140 g for 60 s. Cells were left to incubate for 24 hr at 25°C in a cell culture incubator with no $CO_2$ adjustment. After incubation, plates were opened and 15 µl of 150 nM solution of PD-0325901 (MEK1/2 inhibitor, CAS: 391210-10-9, Cayman Chemical, 15 nM in assay) dissolved in medium with 5% DMSO (Sigma Aldrich, Ref: 41644–1 l, Merck KGaA, Darmstadt, Germany) or 5% DMSO in medium were added using MultiDrop dispensing. Column 1–6 of each plate were filled with the DMSO control medium and columns 7–12 of each plate were treated with the inhibitor solution. Plates were heat sealed again and left to incubate for an additional 48 hr, 72 hr and 96 hr at 25°C without $CO_2$ adjustments prior to cell lysis and RNA extraction.

## Cell lysis and RNA extraction

Cell lysis of 3 plates from each time point was carried out after 48 hr, 72 hr and 96 hr on a Beckman FX pipetting robot using a custom protocol with reagents supplied from the Agencourt RNAdvance tissue kit (Beckman Coulter, Ref: A32649). RNA extraction was performed on a Beckman NX-P robot with a custom protocol using lab ware and the reagents from an Agencourt RNAdvance tissue kit. In brief, medium was removed and the cells were washed carefully using 200 µl PBS before 50 µl freshly prepared lysis buffer were added (5% Proteinase K in 30 ml of lysis buffer). Lysis was left to incubate for 30 min at room temperature (RT). The solution was then mixed, transferred to a new plate and stored at −20°C. Cell lysates were mixed vigorously in a 1:1 mass ratio with the magnetic bead solution (320 µl 70% 2-Propanol + 80 µl magnetic beads) and incubated for 5 min at RT prior to 5 min of incubation at RT on a plate magnet. The supernatant was removed and the plate removed from the magnet. The samples were re-suspended in 800 µl of washing buffer before another 4 min of incubation at RT on the magnet. The remaining supernatant was removed, the plate removed from the magnet and the samples dissolved in 800 µl of 70% ethanol. After gentle mixing of the solution, samples were left to incubate for 4 min at RT on the magnet and the supernatant was removed. Prior to addition of 100 µl DNase mix (80 µl $H_2O$, 10 µl DNase buffer, 10 µl DNase) sample were incubated for 5 min at 37°C on a pre-warmed heating block. Samples with DNase were then incubated another 15 min at 37°C on the heating block before 500 µl of washing buffer were added and the samples were mixed gently by pipetting. Samples were left to rest for 4 min at RT before the plate was transferred to the magnet and incubated for 4 min at RT. The supernatant was discarded and the samples washed three times with 200 µl 70% ethanol. The supernatant was removed and the samples allowed to dry 5 min at 37°C before they were vigorously re-suspended in 30 µl RNase, DNase free water. RNA was allowed to dissolve from the beads by incubating the mixture 5 min at RT before the plate was transferred to the magnet. After 30 s, the DNase treated total RNA containing eluate was transferred to a fresh plate, pooled per sample and stored in conventional Eppendorf tubes at −20°C.

## Sample preparation and sequencing

Sample preparation and sequencing were carried out by the DKFZ high throughput sequencing core facility using standard protocols. Sample libraries were prepared using an

Illumina RNA TruSeq protocol and sequenced on a HiSeq V4 using 125 bp paired end mode. All samples passed quality controls and provided on average 10 million reads to be mapped.

## Fixation, staining and imaging of a plate for visual inspection

A plate for visual inspection of experimental outcome was prepared by fixation and fluorescent staining of cells after 72 hr of post treatment incubation. All steps were carried out manually. The cell culture medium was discarded and the cells were washed twice with 100 µl 1xPBS. The PBS was removed and 100 µl fixation buffer (4% PFA (Roth, Ref: 0335.3, Carl Roth GmbH +Co. KG, Karlsruhe, Germany), 0.3% Triton-X (Sigma Aldrich, Ref: T8787-250ml), 0.1% Tween20 (Sigma Aldrich, Ref: P1379-100ML), 1% BSA (GERBU, Ref: 1507.0100, GERBU Biotechnik Biotechnik GmbH, Heidelberg, Germany) in 1xPBS were added prior to incubation for 45 min at RT in the dark. Prior to another washing with 100 µl PBS, the fixation buffer was removed and after washing the plate was stored filled with 100 µl of PBS. For staining the PBS was removed and the cells blocked by incubating 20 min at RT with 100 µl blocking solution (0.3% Triton-X, 0.1% Tween 20, 4% BSA in 1xPBS). After blocking, the blocking solution was removed and the samples were incubated 30 min at RT in the dark in 50 µl the staining solution (1:4000 Hoechst (Thermo Scientific, Ref: H1399, Life Technologies GmbH, Darmstadt, Germany), 1:1500 primary FITC labelled anti α-tubulin antibody (Sigma, F2168-.5ml), 1:6000 Phalloidin-TRITC conjugate (Sigma, P1951) in 1x blocking solution). The stained cells were washed twice in 50 µl PBS and stored in 50 µl PBS at 4°C. The plate was imaged on an IncellAnalyzer 2200 (GE Healthcare GmbH, Solingen, Germany) automated microscope (Channels: DAPI (ex/em: 390/435) at 400 ms, Cy3 (ex/em: 542/597) at 400 ms, FITC (ex/em: 475/511) at 300 ms). Images were analyzed using the same R/Bioconductor pipeline as is described below.

## Data analysis and statistical testing

Data were analyzed using the DeSeq2 R/Bioconductor package. In short, reads were subjected to FASTQC based quality assessment and all samples with good quality were mapped against the BDGP6 *Drosophila melanogaster* reference genome using the RNA-STAR aligner with pre-set default parameters for paired end RNA sequences. Resulting BAM files were then subjected to normalized feature counting using htseq-tool by *Love et al., 2015*. DeSeq2 was used for normalization of count data and statistical testing of differential expression between different conditions. During the statistical analysis default parameters were chosen as supposed in the methods of *Love et al., 2015*. Quality was assessed by replicate correlation using spearman rank correlation. Fold changes of read counts between conditions were calculated as log2 normalized ratio of quantile normalized read counts between two conditions. Significance of differential expression was assessed using linear mixed effect modelling and corrected for multiple testing by p-value adjustment after Benjamini Hochberg (FDR). Cut-off for significantly dysregulated genes was an FDR of 10%.

## Genome wide HD3 dsRNA library

The HD3 genome wide Drosophila dsRNA library was previously described in *Horn et al. (2010)* and *Billmann et al. (2018)*. It contains 28941 sequence unique dsRNA reagents targeting 14242 unique gene IDs and comprising two sequence independent reagents targeting each gene, where possible. The reagents were optimized for efficient targeting of the most recent (BDGP5) mRNA annotations in *Drosophila melanogaster* by for example avoiding CAN repeats and non-unique sequences (off-targets). Target genes in the library are organized into the categories of chromatin biology, human homologs, non-human homologs, human homologs II and non-human homologs II. The library contains primer pairs against exonic regions of target genes that are amplified by PCR and processed to double stranded RNA by in vitro transcription (IVT) the following way. Starting from purified genomic DNA of S2 cells, a first PCR (0.28 µM fwd./rev. primer, $H_2O$, 10x Buffer (Qiagen, Ref: 203209, Qiagen GmbH, Hilden, Germnay), 0,28 mM dNTPs (Fermentas, Ref: R0182, Life Technologies), 0.03 U Hotstar Taq (Qiagen), 2 ng genomic DNA) with genomic sequence specific primers was performed, which also adds four different TAGs for later re-amplification of the product. Using

TAG specific primers, a second PCR (same conditions as first one) was performed adding an additional T7-promotor sequence to either end of the resulting amplicon. A list of primers can be found at *Supplementary file 4*. The product of this reaction was then input for in vitro transcription at 37°C, incubated overnight (10 µl buffer (200 mM Tris-HCl, 100 mM MgCl$_2$, 50 mM DTT, 10 mM Spermidine), 20 µl NTPs (Sigma), 5 µl Isopentenyl pyrophosphate (IPP, Sigma), 0,25 µl RNA lock (40 U/µl, Fermentas), 5 µl T7 Polymerase (20 U/µl)). The IVT product (dsRNA) was normalized to 250 ng/µl and diluted to a final concentration of 50 ng/µl in DNase, RNase free water. All products were subjected to E-GEL (E-Gel 96 2% 8-PAK, Life Technologies) quality control using the QIAXEL automated gel electrophoresis (Qiagen). Samples, which did not pass quality control (no band, band of the wrong size or smear) were processed again or flagged in the annotation. The final solution was stored in 384-well stock plates and aliquoted to 384 Greiner µ-clear plates prior to the screening assay at a mass of 250 ng/well. A table containing all HD3 library IDs that were used in the combinatorial RNAi screen can be found in *Supplementary file 5*.

## Measuring RNAi and/or drug induced phenotypes by high-content imaging

All the following experiments have been carried out using the same protocols for cell culture, fixation, staining, imaging and image analysis. All reagents were from the same vendors, though LOTs may vary and treatment and incubation times were subject to experimental design. Thus, the general workflow for automated assaying of cellular reactions by a high-content readout will be introduced first followed by a detailed description of the different experiments that were performed.

### dsRNA pre-spotting for high-throughput screens

dsRNA reagents dissolved in water were spotted into barcoded 384-well plates for microscopy (Greiner µClear, black, flat-transparent-bottom, Ref: 781073) to reach a final mass of 250 ng dsRNA in each well (5 µl of a 50 ng/µl solution). Express V medium (Gibco, Ref: 10486–025) with 10% Glutamax (Gibco, Ref: 35050–061) were pre-warmed to 25°C and 30 µl dispensed on top of the spotted dsRNA using a MultiDrop dispenser (high speed, 384-well standard plate, full plate, 20 µl pre-disp. and 16 mm offset).

### Cell seeding and culturing conditions

Cultured S2 cells were detached from cell culture flasks (T175, Greiner, Ref: 12668) by rough rinsing with pre-warmed medium and counted via a Nexcelom Cellometer Auto 1000 using Trypan blue viability staining. 10 µl of pre-diluted S2 cell solution were then seeded to a final concentration of 9000 cells/well into the prepared medium using MultiDrop dispensing (low speed) under constant steering in a sterile Corning spinner flask (half maximal power on Variomag Biosystem magnet stirrer). Cell containing plates were heat sealed using the PlateLoc (Agilent, 2.5 s, 182°C) and centrifuged at 140 g for 60 s. Cells were then left to incubate for 24 hr at 25°C without CO$_2$ adjustment in a Binder cell culture incubator. A culture was kept in stock for maintenance (60 × 10$^6$ cells in 20 ml ExpressFive +10% Glutamax, splitted every 4 days and expanded for assays if needed).

### Compound treatment

After 24 hr of incubation, plates with growing cells were opened again and small molecule treatment was added. The concentration of applied compound is outlined with the separate experiments. Each time 5 µl of a solution containing 5% DMSO (Sigma) in medium, or PD-0325901 dissolved in 5% DMSO in medium, was added to an assay concentration of 0.5% DMSO and varying concentrations of compound. After compound addition, plates were closed again using heat sealing (2.5 s, 182°C) and incubated for another 48 hr to 96 hr depending on the experiment at 25°C without CO$_2$ surveillance.

## Fixation and staining of cells

After treatment, cells were subjected to fixation using a robotics procedure on a CyBiWell Vario with 384-pipette head (Analytic Jena AG, Jena, Germany). There, medium was removed and cells were washed with 50 μl PBS (Sigma Aldrich, Ref: P3813-10PAK). After addition of 40 μl of Fix-Perm solution (4% Para-formaldehyde (Roth, Ref: 0335.3); 0,3% Triton X-100 (Sigma Aldrich, Ref: T8787-250ml); 0,1% Tween20 (Sigma Aldrich, P1379-100ML; 1% BSA (GERBU, Ref: 1507.0100)), plates were left to incubate for 60 min at RT and then washed twice with 50 μl of PBS. 50 μl of PBS were added again and plates could be stored at 4°C prior to staining. Fixed cells were first blocked by adding 30 μl of blocking solution (4% BSA; 0,1% Triton X-100, 0,1% Tween20) and incubated for 30 min at RT. Next, the blocking buffer was removed and 10 μl of staining solution (1:4000 Hoechst (Invitrogen, H1399), 1:1500 primary FITC labelled anti α-tubulin antibody (Sigma, P1951), 1:6000 Phalloidin-TRITC conjugate (Sigma, F2168-.5ml) in 1x blocking buffer) were added. The plates were left for staining for 60 min at RT. After staining, 30 μl of PBS were added and the staining solution was removed. After two additional washing steps in 50 μl PBS, the plates were filled with 50 μl of fresh PBS and stored until imaging.

## Imaging of high-throughput plates on an InCell-Analyzer 2200

All plates were imaged using the same protocol. There, the InCell-Analyzer 2200 (GE-healthcare, hereof short InCell) was calibrated to scan Greiner μClear plates setting the bottom height to 2850 μm and the bottom thickness to 200 μm so that the laser autofocus function can find the bottom of the cells at 10% laser power. This function measures the refraction peaks of the laser while moving the focus in Z-direction. Two peaks are detected, where the second peak marks the position where the focal plane hits the inner side of the well bottom. This Z-position was used for image acquisition in three channels: DAPI (excitation: $390 \pm 18$, emission: $435 \pm 48$) at 400 ms exposure (100 ms in dose response experiments and genome wide screens), Cy3 (ex: $475 \pm 28$, em: $511 \pm 23$) at 300 ms exposure (200 ms in dose response experiments and genome wide screens) and FITC (ex: $542 \pm 27$, em: $597 \pm 45$) at 300 ms exposure (300 ms in dose response experiments and genome wide screens). Four tiles per well were imaged as fields of view each representing a 665.60 μm x 665.60 μm area covered by $2048 \times 2048$ pixels. The fields were centered around the well's center in a layout touching each other's margins. A KinedX robot (PAA Scara, Peak Analysis and Automation Ltd, Hampshire, UK) was used to automate feeding of plates into the microscope. For efficient data handling, the microscope's workstation was directly connected to a server cluster storage node via X520 optical fiber network adapter (Intel).

## Image processing and feature extraction

On a 96 CPU server cluster, a script scheduled the automated processing of image files through the following analysis workflow, representative for each field of each well on each plate. First, the images were read in and, before each image is duplicated, each dye was assigned to the cell organelle it is supposed to stain (DAPI: DNA, Cy3: actin, FITC: α-tubulin). The duplicate image was log2 transformed, scaled from 0 to 1 and smoothened by a Gaussian filter using a sigma of one and a radius of 7 pixels. This reduces optical noise and smoothens the image gradients for further segmentation by thresholding. Afterwards the normalized actin and tubulin images were binarized by global thresholding (thresholds of 0.2 and 0.35, respectively). If the tubulin channel was bright enough (*diff (range (raw image tubulin)) > 0.1*) the binary cell body mask was defined as the sum of the actin and tubulin binary masks, else the tubulin was left out of the cell body segmentation. Second, the cell nuclei were identified by applying a local adaptive average threshold to binarize the nuclei image and assigning objects after morphological smoothing of the binary image. Each normalized and smoothened DAPI channel image was binarized using an average thresholding method with a $21 \times 21$ pixel wide filter. Every Pixel above the local average plus the given offset is defined as foreground and every other pixel is defined as background. The resulting binary image was then subjected to morphological operations of opening and hull filling such that filled objects with smoothly roundish outlines result. The function *bwlabel* numbers each area of connected foreground pixels to be an object (here nuclei). Offsets for segmentation were varied if the channels

surpassed certain thresholds. If more than 30 nuclei were counted per field, the objects were subjected to further propagation of nuclei objects into the *á priori* defined cell body mask. Starting from the nucleus objects as seed regions, the cell bodies are segmented by propagating the nuclei objects into foreground area. Allowed foreground area is defined by the cell body binary mask. The direction of propagation is defined by the image gradient of the normalized actin channel image, together with the lambda factor. The extend of lambda defines how strong the image gradient is considered. If lambda is one the half distance between two nuclei defines the cell border and the cell bodies equal a Voronoi map between the nuclei. Here a relatively small lambda was chosen to allow also for cell shapes other than roundish cells. There, the lambda factor defines to which extend the propagation algorithm follows the image gradient of the normalized actin image while the body binary provides the borders of the propagation. Using the segmented object outlines as masks on the original image, cell features have been extracted using EBImage's *computeFeatures* function on each object and original non-normalized channel of the image. Specifically numeric descriptors for four feature classes are defined in this function (**Supplementary file 6**): (i) shape features describing how the object outline forms, (ii) basic features that describe pixel intensity based summary statistics, such as 5% quantiles, of pixel intensity within the borders of the object, (iii) moment features that describe the spatial orientation of the objects, (iv) Haralick features derived from a pixel intensity co-occurrence matrix as texture descriptors and (v) social features such as distance to the first 20 nearest neighboring cells. Cell body shape and moment features (dependent on the cell outlines) were only computed once using the actin channel. The tubulin channel, because of having the same outlines as the actin channel, was only measured for basic, intensity and texture features. All categories giving the nuclear shape, moment, intensity and texture features were measured in the DAPI channel. Social features were derived by a k-nearest-neighbor search based on the geometric centers of segmented nuclei. Taken together, we extracted 78 features for every single cell. Single cell data were stored and aggregated to well averaged data by calculating the trimmed mean (q = 0.01) of all cells belonging to all fields of one well and its standard deviation.

### Genome-wide chemo-genetic interaction screen

We performed genome-wide RNAi screens in combination with differential drug treatment to test dsRNA reagents, identify interesting candidates for combinatorial screening and to find which genes react most differentially as resistance mediators or sensitizers to the MEK inhibitor (PD-0325901) treatment.

## Genome wide high content RNAi vs drug screen

Four sets of 88 × 384 well Greiner µClear plates were spotted with the HD3 library, 5 µl of 50 ng/µl dsRNA in each well. This library targets one gene with one dsRNA per well. Most genes were targeted with two separate designs per gene, where possible. Each plate contained the same controls. Two additional plates, containing only controls were added for controlling assay stability. Controls were chosen to spread over the complete dynamic range of cell fitness. dsRNAs against *RLUC* and *GFP* expressing plasmids serve as non-targeting negative controls, such that we could control for unspecific dsRNA induced phenotypes.

## Assay workflow

Briefly, dsRNA containing plates were thawed, seeded with cells and left 24 hr for incubation prior to drug treatment. Plates were opened and 5 µl of 15 nM PD-0325901 in 5% DMSO were added resulting in a final concentration of 1.5 nM PD-0325901 in 0.5% DMSO in medium using a MultiDrop dispenser. The cells were left to incubate for another 72 hr at 25°C prior to fixation, staining and imaging. Images were acquired using the standard protocol with low illumination timings (DAPI: 100 ms, Cy3: 200 ms, FITC: 300 ms). The resulting images were analyzed in line with the acquisition using the standard image analysis pipeline.

## Data analysis workflow

Resulting data were collected to a data frame containing feature values aggregated per well as trimmed mean and standard deviation. This data frame was then reformatted to a 4-

dimensional data cube featuring the dimensions: feature, plate, well, screen. Each feature's minimum was added to the feature prior to $\log_n(x + 1)$ transforming to adjust each feature's histogram to a more normal distribution. Following transformation, each plate in each screen was normalized separately for each feature by B-Score normalization. There, a median polish was performed across the plate to remove spatial column and row biases from the plate. Additionally, the B-Score normalization centers and scales the data to be the residuals of the median polish divided by the median absolute deviation (mad) across all values of the plate and thus be symmetrically centered around zero and scaled in units of mad. An artifact resulting from the dispensing pattern of the MultiDrop dispenser is a Chess-board like pattern biasing all values spatially. Under the assumption that the values of the black panels and the white panels should both follow the same distribution, we performed the following normalization. We subtract the absolute difference of the means of the two groups from the group with the higher mean, thus equaling the two group means $(X_{high,1} = X_{high,0} - |mean(X_{high}) - mean(X_{low})|)$. This ensures that all features, plates and screens are on the same normal scale and can thus be compared statistically. Here, 38 representative features were chosen manually based on their biological significance (our ability to refer them back to cellular phenotypes) and their technical reproducibility between the two mock (DMSO) treated replicate screens and their information content, as measured by added variance (**Supplementary file 6**). From the screening data, we derived metrics to judge whether a gene is a suitable candidate for gene-gene-drug combinatorial screening. For this

purpose, several metrics have been deployed based on the Euclidean distances ($\sqrt{\sum_{i=1}^{n}(x_i - y_i)^2}$)

between samples and controls under different conditions, Pearson correlation coefficients between independent design replicates and effect size compared to control. They are described comprehensively in the methods part and further summarized in **Supplementary file 1**.

## Combinatorial RNAi screening under differential time and treatment conditions

During this work, we conducted an experiment were genes were knocked down in a pairwise manner and under differential treatment conditions. After treatment, the process of phenotype development was followed over time. All lab ware and reagents, which are not further detailed here, were equivalent to previous experiments. The dsRNA sequences that were used in the combinatorial library can be found in **Supplementary file 5**.

### Gene combinatorial library design and synthesis

168 genes were chosen for combinatorial library design. The library contained 12 batches for screening, each comprising 80 384-well Greiner μ-clear plates spotted with 250 ng dsRNA/well dissolved in 5 μl of DNase, RNase free water. dsRNAs were obtained from the HD3-library templates and synthesized accordingly. To avoid contaminations, all dsRNAs were sterile filtered using Steriflips-0.22μm for the query dsRNAs and MultiScreenHTS-GV 0.22 μm filter plates for the target dsRNAs. Here, genes were divided into target and query genes and screened a matrix of 76 genes combined with 168 genes. We defined the larger group as target genes and chose 76 of them to be the query genes. We screened each target gene in two sequence independent designs and each query gene in one design. This way, we screened 25536 dsRNA combinations (12768 gene pairs) in each batch. Combinatorial dsRNA spotting was achieved with combining the query and target master plates such that 2.5 μl of each query dsRNA were spotted onto 2.5 μl target dsRNA by a Beckman FX liquid handling robot using a 384-well pipetting head. This results in 76 × 384 well plates where each well, except the control wells in column 12 and 13, contains 250 ng of dsRNA targeting two different genes. Controls in columns 12 and 13 contained only 125 ng of dsRNA in the same volume of water. Control plates containing the uncombined target gene dsRNA matrix with 250 ng dsRNA per well complete one screening batch of 80 plates. To achieve differential treatment and time resolution, 12 screening batches were prepared. They were divided into two groups of six batches, which then were treated under different conditions in duplicate. six

library batches are needed to screen two conditions (1.5 nM PD-0325901% and 0.5% DMSO) at three time points (48 hr, 72 hr, 96 hr), all in all comprising 480 screened plates. The entire experiment was repeated twice and we screened 960 × 384 well plates.

## Assay workflow

The assay workflow follows the same procedures as outlined above. Briefly, 9000 cells per well were seeded onto 384-well Greiner µClear plates for microscopy, which were pre-spotted with a combinatorial dsRNA library. After centrifugation, plates were sealed and left to incubate for 24 hr at 25°C prior to compound addition. Therefore, a PD-0325901 solution, (15 nM in medium with 5% DMSO), and a 5% DMSO only dilution in medium were prepared and added to the opened plates. This resulted in either 1.5 nM PD-0325901 or 0.5% DMSO in-assay concentrations. Plates were closed again by heat sealing and left to incubate until the experiment was stopped by fixation after 48 hr, 72 hr and 96 hr, respectively. Fixation with 4% PFA and staining with Hoechst, Phalloidin-TRITC and anti-α-tubulin-FITC conjugate antibody were carried out using standard protocols on a CybiWell Vario liquid handling workstation. Stained plates were imaged on an InCell Analyzer 2200 automated high-throughput microscope according to the protocol above with 20x magnification, three channels per field and four fields per well.

## Data analysis workflow

Resulting images were analyzed using the R/EBImage based pipeline described above. As a result, we obtained a data frame containing one row for each well that was screened and columns containing the measured features and well, plate and batch identifier for later trace back. The resulting raw data contained 368640 rows and 164 columns. To this end, data acquired from single cells was aggregated by calculating the trimmed mean (q = 0.01) for each feature extracted in the respective well together with its standard deviation. This way, outliers, produced by over or under segmentation of cells, should be excluded from further analysis. Here, data were either kept as a table or transformed into a multidimensional data cube depending on the analysis. Data were normalized by dividing each feature in each plate by the median of the negative control wells (if that was not zero). Further, the values of each feature were glog transformed with c being the 3% quantile of the features value distribution over all values. Glog transformation works effectively in smoothening the histogram of each feature without the anomaly observed by conventional log transformation. For each feature, data were then scaled and centered around 0 by using the robust Z-transform, where the feature median is subtracted from each value and the result was divided by the median absolute deviation (x=(x-med(x))/mad(x)). After that, all feature data were in normalized units of median absolute deviation from the median of that feature normalized to each plate. The normalized feature vector provides the basis to all further analyses. Screening quality was assessed per plate such that plates with a replicate Pearson correlation smaller than 0.6 and a Z'-factor (between Diap1 and RasGAP1) smaller than 0.3 were masked by the NA_real_ constant, so all experimental dimensions remain complete while the values are masked from downstream analyses.

## Statistical Methods

From normalized features per well, the data frame is reformatted to fit the experimental design into a 5-dimensional data cube. The dimensions are: target dsRNA (two entries for each gene), query gene, time, treatment and feature. The data cube is further subjected to genetic interaction inference following the protocol established by *Fischer et al., 2015*. There, genetic interactions are defined as the residuals of a modified median polish over the double perturbation matrix of one replicate, feature, treatment and time point. In the median polish a robust linear fit ($\delta_{ij} = m_i + n_j + \pi_{ij} + \varepsilon$) is used to lift the main effect (m, n) of each query such that it resembles the value of a single gene knockdown. The residuals of this fit scaled by their median absolute deviation are then defined as $\pi$-scores. $\pi$-scores further provide us with a quantitative measure of genetic interaction following the multiplicative model plus an error term ($\varepsilon$) estimating the experimental noise. There, the interaction of two genes is defined as the deviation of the measured combined phenotype ($\delta$) from the expected. The expected

phenotype would then be a multiple of the two independent single knockdowns. The resulting $\pi$-scores were collected for all replicates (dsRNA and experimental) and the significance of their mean over all measured scores was estimated by a moderated students t-test as is implemented in the R-package limma. There, the t-test was adapted for situations where a small amount of observations was tested in many tests, normally causing large test variability, using an empirical Bayes variance estimator. P-values were adjusted for multiple testing using the methods of Benjamini Hochberg method. From there on, adjusted p-values were used to control false discovery rates. The FDR estimates the chance that the finding was observed by chance given the entire dataset.

## qPCR analysis

Quantitative real time PCR (qPCR) was used to analyze the transcriptional response following Rel/pnt co-RNAi. To this end, as $5*10^5$ cells / well were seeded in 630 µl ExpressFive (Gibco) culture medium (supplemented with 10% Glutamax, Gibco) and reverse transfected with 14 µg dsRNA (0,2 µg/µl solution in sterile DNase/RNase free water), at the time of seeding, without transfection reagents. For this experiment, all dsRNAs denoted with #2 were used in three biological replicates. Combinatorial RNAi was achieved by mixing 7 µg of dsRNA targeting each gene (**Supplementary file 8**). After 72 hr incubation at (25°C), cells were washed once in 750 µl PBS (Gibco) and lysed in 350 µl RLT buffer shipped with the RNAeasy-mini Kit (Qiagen). RNA was then purified from all samples according to manufacturer's standard instructions for spin column purification. An optional DNase digestion step was performed using the RNase-Free DNase Set (Qiagen). RNA was diluted from the spin columns using 40 µl RNase/DNase-free water. The concentration was measured using a NanaDrop spectrometer (Thermo scientific). Samples were prepared for qPCR by reverse transcription of 1 µg of RNA using RevertAid H minus First strand cDNA Synthesis kit (Thermo scientific) according the manufactures standard protocol. A qPCR reaction was prepared using PrimaQuant 2x qPCR-Mastermix (Steinbrenner, Steinbrenner Laborsysteme GmbH, Wiesenbach, Germany) by mixing 5 µl of sample (1:10 diluted cDNA) with 5 µl of Mastermix (including 0,3 µM of forward and reverse primer) on a 384-well qPCR plate (LightCycler 480 Multiwell Plate 384, white, Roche, Ref: 4729749001, Roche Molecular Systems Inc., Pleasanton, California, USA). The plate was then centrifuged (2 min, 2000 rpm) and processed for qPCR in a Roche 480 LightCycler using the following PCR program: (i) 10 min at 95°C, (ii) 15 s at 95°C, (iii) 60 s at 60°C, repeat step ii and iii 40 times and measure fluorescence at 494 nm-521 nm during step. Melting curve analysis of each sample was performed to assess reaction quality. Relative expression of each gene in each sample (normalized to rps7 expression) was analyzed as $\log_2$-foldchange over RLUC dsRNA treated samples. qPCR primers were designed using the GETprime web service (Supplementary file 9).

## qPCR and Western Blot analysis of Rel/pnt crosstalk

Since phosphorylation levels of rl were already equalized after 96 hr of different dsRNA treatments (*Sos* k.d., MEK inhibition and rl k.d. showed no or little reduction in pp-rl, data not shown) we examined the molecular response also 48 hr after dsRNA transfection. To this end, we seeded $1*10^6$ S2 cells per well into a 12-well cell culture plate (Greiner) and transfected them with 28 µg of each dsRNA (14 µg + 14 µg in case of co-RNAi). Cells were then harvested after 48 hr and proceeded to qPCR analysis as outlined above. In this experiment sequence, independent dsRNAs targeting each gene were used (#1 and #2 of each gene) as replicates. Phenotypic analysis during image-based screenings confirmed high reproducibility (PCC >0.8) between either dsRNA reagents targeting the same gene. Another replicate of these same samples was proceeded to Western Blot analysis of pp-rl levels according to the same protocol as outlined previously.

