## [Decision Letter]

[**Editorial note:** This article has been through an editorial process in which the authors decide how to respond to the issues raised during peer review. The Reviewing Editor's assessment is that all the issues have been addressed.]

Thank you for submitting your article "Time-resolved mapping of genetic interactions to model rewiring of signaling pathways" for consideration by *eLife*. Your article has been reviewed by three peer reviewers, and the evaluation has been overseen by a Reviewing Editor and Aviv Regev as the Senior Editor. The reviewers have opted to remain anonymous.

The Reviewing Editor has highlighted the concerns that require revision and/or responses, and we have included the separate reviews below for your consideration. If you have any questions, please do not hesitate to contact us.

In this study, Boutros and colleagues study the plasticity of genetic interactions by performing an imaging based RNAi screen using combinatorial knockdowns at multiple timepoints in *Drosophila*. By studying in particular on the RAS signaling pathway, authors showcase several examples of context dependent crosstalk. All three reviewers are interested in the study and thought that this will be an interesting resource for the community.

However, concerns were raised regarding the clarity of methodology to make it accessible and useful resource for the field. Also, the manuscript currently appears very technically written and therefore reviewers have made several suggestions for sections where optimization is required so that the main message and importance of the study comes out more clearly.

Separate reviews (please respond to each point):

*Reviewer #1:*

This is an extremely technically and statistically thorough study using quantitative phenotyping in *Drosophila* cells to look at genetic interactions in sub-genome scale set of genes. WHat marks this out in particular is that the authors are specifically interested in interactions that change over time following chemical perturbations. In broad terms this is a really important question and the thoroughness of both the experimental approach and the rigour of the statistical framework that they devise to examine and compare GIs is outstanding. I am thus happy both with the importance of the science and the technical quality of the work.

My main comments would be that the authors need to flesh out some of their data descriptions so that we know more why these conclusions are important and what they mean biologically as well as what happened. This is most true (for me at least) in the section 'Differential GIs enrich in stress responsive genes…".

1) ksr and rl form a syn sick interaction independent of Ras – what does this mean biologically? How does it change how we think they act?

2) Next paragraph skd and Stat92E – synlet connects them until perturbed. Same question – how does this affect our view of how these pathways work?

3) High numbers of alleviating (nuclear shape) or aggravating (nuclear texture) stable interactions. Is this different to what we expect? Why? What is the null expectation for these results?

4) Exceptionally high time dependence for interactions with housekeeping genes. Why?

GIs are pretty abstract entities. How they are expected to change across time is something I cannot guess so having some idea what we might have expected and what these findings reveal would really help.

I'd also suggest in future that as well as focusing on a well chosen gene set as they do here, they also do similar analyses on a set of random genes so we can see whether the trends they find are specific to the chosen genes or to genes in general. I know that adds greatly to the work (hence not requiring it here) but I think it would be useful.

Minor Comments:

Results section, paragraph twenty; extent

Figure 2 – I'd prefer something other than yellow in A. It's faint on the page.

*Reviewer #2:*

In this work, Heigwer et al. develop a method to map genetic interactions in a time-resolved manner. They use automated microscopy of *Drosophila* cells treated with dsRNAs and/or chemical compounds to study "trigenic" interactions.

This is mainly a technology paper, with some quite interesting preliminary observations. The main fleshed out contribution here is the time-resolved analysis and associated tools. The manuscript is clearly (albeit very verbosely and technically) written, and should be published after corrections below.

Major:

– There is excessive technical and numerical detail in the main text, making the work difficult to access. This will limit the ability of general readership to discern the main contributions. I would recommend moving some text to methods section, and more technical figure panels to the supplement, and making a new final model figure that summarizes the main contributions and highlight the rationale of using the time-resolved approach. I would try to answer the question: What did we find because we used this technology that we could not have found using a more standard end-point genetic interaction analysis?

– The outputs that are studied are expected to behave differently over time. Cell number expands exponentially, whereas cytoplasmic phenotypes could be largely additive or have sigmoidal response functions. The type of "interaction" detected will depend on the assumed shape of the downstream input-output curve. This should be discussed, possibly citing relevant papers such as that from the Lehner group in recent *eLife*. It should also be considered whether the different temporal responses of interactions between components involved in distinct biological processes result from differences in shapes of the input-output curves.

– Feedback will also lead to epistasis, and the authors' approach might be able to detect the mechanisms of feedback, as transcriptional feedback is expected to be slower than phosphorylation- or allostery-based feedback, and translational or transcriptional components might behave differentially when they are required for feedback. Analysis of this aspect would increase the impact of the study and give the readers a sense of "why". Now the paper looks a bit like "we did this because we can".

Minor Comments:

Abstract "end-point assay"

Results paragraph seven; "a control perturbation"

Paragraph two subsection “Differential genetic interactions enrich in stress responsive genes and pathways”; "to such an extent that"

*Reviewer #3:*

In this manuscript Heigwer and colleagues study how multi-parametric genetic interactions inferred from double knock-down experiments change during time in the presence or absence of a kinase inhibitor. Pair-wise gene deletions or knock-downs have been used for a long time to study how the consequence of the double gene perturbations to a phenotype is different from expectation. More recently, such pairwise combinations have been performed in different environmental conditions or in different genetic backgrounds. In this study the authors asked how such genetic-interactions change during time. Time dependencies in genetic-interactions were previously exploited but not studied, as mentioned by the authors, by Shen and colleagues (Shen et al., 2017). For this purpose they have developed a linear regression model to jointly estimate the impact of the kinase inhibitor and the time dependencies of the genetic-interactions. The authors go to great lengths to set-up the experiments and to show the validity of their data and approach. In fact, about half of the results are devoted to this. They first performed a single gene down-regulation screen in the presence/absence of MEK inhibitor using microscopy and imaging as cellular phenotypes. Based on this they selected a set of candidate genes to perform a 76 by 168 double gene knock-down experiments in 3 time points (+/- MEK inhibitor). They performed a series of benchmarks on the screens to ensure that the data is reproducible.

From the genetic-interactions screens there are some general analysis and findings. From a technical point of view the authors show that modelling the time dependencies as they do can identify a higher number of differential interactions when compared to previous approaches. Biologically, most of the interactions with changes over time were interactions that changed in magnitude over time in a way that was similar in the control and MEK inhibitor (called "stable" by the authors). Those interactions that changed over time differently in the inhibitor versus the control were found to be enriched in signalling genes. One aspect that I found particularly interesting was the possibility to assign a magnitude of time dependence to each differential interaction. Based on this they briefly explored how some phenotypes and some genes were more responsive to changes after perturbation. The authors showed that the correlations of the changes interactions is predictive of functional association. Finally, from the analysis of correlation of differential interactions, they then selected and further investigated the role of Rel as a potential negative regulator of Ras signalling.

Overall, I think this work explores well the changes in genetic-interactions over time after perturbation. The analysis is well performed and extensively benchmarked. I have the impression that the manuscript focuses almost more on the generation of the data than on the new insights. It was also not very easy to follow but that is also due to the complexity of the experiment – non linear effects of pairs of gene knock-downs over time plus and minus drug. Even for researchers used to thinking about genetic interactions the results of these assays should be difficult to conceptualize. In part, this complexity also makes it difficult to extract novel findings on the genetic architecture of the cell.

I have no major concerns regarding the work, only a few minor points on clarification that the authors may consider:

– From my reading of the manuscript the major novel findings of this work relate to the time dependencies on the differential genetic interactions. However, almost half of the Results section relate to the process of obtaining the differential interactions. While I commend the authors for the thorough characterization of the chemical-genetic and genetic-interaction data it would be possible to summarize further the initial results, in particular the initial screen for MEK inhibitor sensitive genes and to a lesser extent the establishment of the genetic interactions. This could give more space to focus on the main novel findings regarding the differential genetic interactions. The section on the differential interactions could in turn be broken down into more sub-sections trying to give more visibility to the main new findings that derive from this data.

– The article gets confusing in the nomenclature of the genetic-interactions. These interactions can be: time (in)dependent, differential, stable, positive, negative, aggravating and alleviating. In particular, calling a time dependent interaction as "stable" is awkward. Perhaps using condition independent or condition insensitive instead of stable would be clearer.

– The differential genetic interaction method is not described in the Materials and methods section. Are all genetic interaction pairs tested for differential interactions ? Can a genetic interaction be called as significantly changing without having a strong effect size in any condition? Looking at the genetic interaction scores of replicates in Figure 2—figure supplement 5, the values range from around -5 to 5 with strong variance around -0.5 to 0.5. However, in several places in the article the interaction values and changes highlighted are often in the ranges of 0 to 0.5. Are these interaction terms calculated in the same way in Figure 2—figure supplement 5? It would appear that the variance in replicates is higher than the apparent precision of the method.

– How are condition independent interactions ("stable") defined ? A cut-off on the condition sensitivity parameter ?

– Given that the differential interactions are being called with a linear model on the time differences with a term for the drug effect, are time invariant but drug dependent genetic interactions well captured by this model ?

Minor Comments:

- Figure 1E – annotate in the figure what are the Ras pathway components and translational regulators groups described in the Results section for clarity.

- Figure 4F – could be move to supplementary materials and corresponding results description shortened.

- Figure 2—figure supplement 2 – figure title is wrong

---

## [Author Response]

[…] By studying in particular on the RAS signaling pathway, authors showcase several examples of context dependent crosstalk. All three reviewers are interested in the study and thought that this will be an interesting resource for the community.However, concerns were raised regarding the clarity of methodology to make it accessible and useful resource for the field. Also, the manuscript currently appears very technically written and therefore reviewers have made several suggestions for sections where optimization is required so that the main message and importance of the study comes out more clearly. We would therefore like you to take the reviewers suggestions into consideration and send us a revised manuscript back within two months.

We thank the editor for the comments. We especially focused our revisions on improvements of the accessibility of our concepts and findings to a broader scientific audience. We shifted more technical sections to the methods or supplements, improved consistent definitions of context dependent genetic interactions and highlighted the biological relevance of our observations using a number of examples from our data set.

Separate reviews (please respond to each point):

Reviewer #1:

This is an extremely technically and statistically thorough study using quantitative phenotyping in Drosophila cells to look at genetic interactions in sub-genome scale set of genes. WHat marks this out in particular is that the authors are specifically interested in interactions that change over time following chemical perturbations. In broad terms this is a really important question and the thoroughness of both the experimental approach and the rigour of the statistical framework that they devise to examine and compare GIs is outstanding. I am thus happy both with the importance of the science and the technical quality of the work.

We thank the reviewer for the comment.

My main comments would be that the authors need to flesh out some of their data descriptions so that we know more why these conclusions are important and what they mean biologically as well as what happened. This is most true (for me at least) in the section 'Differential GIs enrich in stress responsive genes…".

We addressed these suggestions in detail as outlined in the following bullets. We specifically improved the structure of the “differential interaction” section to provide readers a more structured access to the assumptions, data and following implications. We also believe that our data likely contains a much larger number of interesting observations. We however decided to focus on a few specific examples to improve the accessibility of our manuscript. We thus want to note that we did not extend the number of specific examples but rather kept the focus on few well-chosen examples and highlighted their biological implications. The examples are chosen to provide the readers an easy access to the complex concepts covered by the manuscript.

1) ksr and rl form a syn sick interaction independent of Ras – what does this mean biologically? How does it change how we think they act?

To follow this suggestion we added a more detailed, yet concise, discussion about the potential impact of our observations on the way *ksr* and *rl* interact (subsection “Examples of context-dependent genetic interactions” and Discussion paragraph three). As such it is reported that ksr serves as a scaffolding protein that supports the phosphorylation cascade downstream of Raf. Whether rl binds this complex is less well studied and no clear evidence is reported. However, our data shows a negative, synthetic sick, interaction which suggest a positive influence on cell viability/proliferation between *ksr* and *rl* independent of MEK kinase activity. This interaction is yet unexplored but could be a potentially interesting target for interventions when kinase inhibitors loose efficacy.

2) Next paragraph skd and Stat92E – synlet connects them until perturbed. Same question – how does this affect our view of how these pathways work?

To answer this question we added a detailed paragraph discussing the triangular crosstalk between STAT, mediator and Ras signaling (subsection “Examples of context-dependent genetic interactions” and Discussion paragraph two). Interestingly there have been a very limited number of studies that suggest cooperative regulatory function of mediator and STAT and others suggest mutual biochemical modulation between STAT and Ras signaling downstream of *Drosophila PDGFR (Tor*). Our study does on the one hand support prior findings regarding the known pathway crosstalk and for the first time demonstrates the linkage of all three pathways. This provides ground for further studies to gain new insights into multi-pathway crosstalk involving biochemical signaling perturbation and transcriptional feedback.

3) High numbers of alleviating (nuclear shape) or aggravating (nuclear texture) stable interactions. Is this different to what we expect? Why? What is the null expectation for these results?

Indeed, we expected that all features show balanced numbers of interaction types. Our data, however suggests that different features have different capabilities to reveal, e.g. alleviating, aggravating, early or late interactions as outlined throughout the manuscript. A possible explanation would be that different biological processes influence different cellular features, e.g. perturbations of the cytoskeleton organization mostly influence shape features while perturbation of nuclear factor alters mostly nuclear texture. The direction of interactions then follows the genes that are involved and so do the different features enrich distinct interaction types. From these observations we concluded that the assessment of multiple features simultaneously gives us a more differentiated characterization of genetic interactions. Our data does not suggest to favor certain phenotypic features over the other as they capture widely different aspects of the cellular reactions. We addressed this comment by re-phrasing the according section in the main manuscript (subsection “Different phenotypic features capture distinct cellular reactions”).

4) Exceptionally high time dependence for interactions with housekeeping genes. Why?

Our data suggests that this phenomenon could be explained if we assume that regulatory networks that are built of housekeeping genes show increased resilience towards perturbation and rewiring. Co-RNAi perturbation as well as chemical inhibition of MEK force the signaling network to adapt its function to maintain homeostasis. If we assume that housekeeping processes are less prone to rapid modulation and need longer time to rewire their interaction networks this would explain our observations. They show exceptionally low treatment sensitivity as their function is so basally governing cellular reactions that perturbation of a downstream signaling pathway does not change their network. Their interactions to other genes also prove this stable that we cannot observe it during early phases of the experiment but very strongly later on. This manifests in their exceptionally high time dependency. We outline the capabilities of our data to map the temporal dynamics these networks in paragraph three of the Discussion section.

GIs are pretty abstract entities. How they are expected to change across time is something I cannot guess so having some idea what we might have expected and what these findings reveal would really help.

To address this comment we completely reworked the end of the introduction and the part of the results which covers Figure 1 and 2. Therein we outline our hypothesis that the quantitative measurement of genetic interactions is always time dependent and explain how we expect this will extent the theoretical framework under which we assess genetic interactions and their context dependence. We further explain that we expect to learn more about the mechanisms by which gene regulatory networks rewire in response to changes in the cell’s environment. From this we can map the plasticity of the gene-regulatory landscape and understand higher-order crosstalk between signaling pathways with possible implications understanding cellular homeostasis and disease.

I'd also suggest in future that as well as focusing on a well chosen gene set as they do here, they also do similar analyses on a set of random genes so we can see whether the trends they find are specific to the chosen genes or to genes in general. I know that adds greatly to the work (hence not requiring it here) but I think it would be useful.

We agree with this comment and consider this a starting point for future studies that are informed by our approach. However due to constraints given by the bad scalability of such complex high-order perturbation experiments we consider an experiment on random genes out of scope of the current study. However, as the degree of automation within lab routines extends, we are certain that such experiments will soon become feasible.

Minor Comments:Results section, paragraph twenty; extent

We corrected this mistake throughout the manuscript.

Figure 2 – I'd prefer something other than yellow in A. It's faint on the page.

We thank the reviewer for this suggestion and replaced the yellow representing alleviating genetic interactions by a less faint orange.

Reviewer #2:

In this work, Heigwer et al. develop a method to map genetic interactions in a time-resolved manner. They use automated microscopy of Drosophila cells treated with dsRNAs and/or chemical compounds to study "trigenic" interactions.This is mainly a technology paper, with some quite interesting preliminary observations. The main fleshed out contribution here is the time-resolved analysis and associated tools. The manuscript is clearly (albeit very verbosely and technically) written, and should be published after corrections below.Major:– There is excessive technical and numerical detail in the main text, making the work difficult to access. This will limit the ability of general readership to discern the main contributions. I would recommend moving some text to Materials and methods section, and more technical figure panels to the supplement, and making a new final model figure that summarizes the main contributions and highlight the rationale of using the time-resolved approach. I would try to answer the question: What did we find because we used this technology that we could not have found using a more standard end-point genetic interaction analysis?

We thank the reviewer these suggestions. We do agree that our manuscript is very verbose on the technical details of the screening methodology and its quality controls. However, we are convinced that this level of detail and thoroughness of quality controls is warranted in such a complex study. In our revised manuscript we performed a number of measures (moving more figures to the supplement, editing Figure 1 and 2, 4 and 5) to increase the accessibility of our manuscript. In paragraph three of the Discussion section we also highlight why the time-resolved measurements allowed to study many observations that were not visible in end-point assays. One of which is the interaction between *Relish* and *pointed* that we followed up in Figure 8. In addition, we shortened the result texts describing initial experiments and shifted details on genome wide screening to the Materials and methods section.

– The outputs that are studied are expected to behave differently over time. Cell number expands exponentially, whereas cytoplasmic phenotypes could be largely additive or have sigmoidal response functions. The type of "interaction" detected will depend on the assumed shape of the downstream input-output curve. This should be discussed, possibly citing relevant papers such as that from the Lehner group in recent eLife. It should also be considered whether the different temporal responses of interactions between components involved in distinct biological processes result from differences in shapes of the input-output curves.

We thank the reviewer for pointing this out. To address this comment, we revised the discussion of modeling the time dependent behavior of different features in the main text (subsections “Robust linear models accurately describe the temporal dynamics of genetic interactions” and “Examples of context-dependent genetic interactions”). Indeed, we expected different signal curves for different features. However due to first, normalization, centering and scaling and second the interaction calling these differences are mitigated when modeling, as also explained by Diss and Lehner, 2018. For example: Two genes that cause a phenotype that follows a sigmoidal shaped function over time. Then this shape is removed together with the main phenotypes of each gene alone when interaction scores are quantified for each time point. The interactions could indeed still follow the curve of the corresponding input feature, however as our study showed as an unexpected result, the remaining residuals could be explained by a linear model, for all features we tested (Figure 3—figure supplement 1). Still as more data become available and more fine-grained time courses and experimental setups are realized, this question could be further investigated.

– Feedback will also lead to epistasis, and the authors' approach might be able to detect the mechanisms of feedback, as transcriptional feedback is expected to be slower than phosphorylation- or allostery-based feedback, and translational or transcriptional components might behave differentially when they are required for feedback. Analysis of this aspect would increase the impact of the study and give the readers a sense of "why". Now the paper looks a bit like "we did this because we can".

Discerning the different scales at which feedback in signaling transduction networks takes place is certainly a very interesting analysis to pursue. To this end, one would need an experiment such as the following: First one creates double knockdown or double mutant cells, second these cells would be pulsed using a chemical perturbation for different time periods. So far this is not different to the experiment we conducted. However, to discern biochemical from transcriptional and later translational feedback one would need an array of time resolved readouts. These would cover overlapping time scales and different biochemical modalities. Therefore, one could use proteomics, very shortly (minutes to hours) after perturbation to map biochemical reactions, transcriptomics to map transcriptional feedback at medium to long time frames (hours to days) and phenomics to assess long-term impact of feedback mechanisms, a mixture of transcriptional and biochemical feedback. Our current dataset however only covers feedback reactions measured a later time points by macroscopic phenotypic changes. Some preliminary data during our follow-up of the *Rel-pnt* interaction indicate a concerted cascade of biochemical and transcriptional regulation that starts early after perturbation and manifests in long lasting transcriptional changes (Figure 8—figure supplement 3 and 4). While one can infer feedback loops from these data and gain insights into their temporal dynamics (e.g. rewiring of housekeeping networks vs. signal transduction networks) we doubt it would have the resolution to exactly discern the mechanisms of different forms of feedback interplay. As also gold-standard data and methods for such analyses are missing, we consider these highly interesting experiments and analyses out of scope of the current study but they offer a very promising avenue to pursue in the future.

Minor Comments:Abstract "end-point assay"Results paragraph seven; "a control perturbation"Paragraph two subsection “Differential genetic interactions enrich in stress responsive genes and pathways”; "to such an extent that"

We thank the reviewer for the suggestions and made changes to the manuscript accordingly.

Reviewer #3:

[…] The authors go to great lengths to set-up the experiments and to show the validity of their data and approach. In fact, about half of the results are devoted to this. They first performed a single gene down-regulation screen in the presence/absence of MEK inhibitor using microscopy and imaging as cellular phenotypes. Based on this they selected a set of candidate genes to perform a 76 by 168 double gene knock-down experiments in 3 time points (+/- MEK inhibitor). They performed a series of benchmarks on the screens to ensure that the data is reproducible.

In the revised version of our manuscript we took the reviewer comment into account and moved several figure panels to the supplement. We further modified corresponding text passages to be more concise. We hope that the Results sections now contain a minimum amount of information required for a thorough assessment of data quality which we believe is of great importance to support claims which are made in downstream analyses.

From the genetic-interactions screens there are some general analysis and findings. From a technical point of view the authors show that modelling the time dependencies as they do can identify a higher number of differential interactions when compared to previous approaches. Biologically, most of the interactions with changes over time were interactions that changed in magnitude over time in a way that was similar in the control and MEK inhibitor (called "stable" by the authors). Those interactions that changed over time differently in the inhibitor versus the control were found to be enriched in signalling genes. One aspect that I found particularly interesting was the possibility to assign a magnitude of time dependence to each differential interaction. Based on this they briefly explored how some phenotypes and some genes were more responsive to changes after perturbation. The authors showed that the correlations of the changes interactions is predictive of functional association. Finally, from the analysis of correlation of differential interactions, they then selected and further investigated the role of Rel as a potential negative regulator of Ras signalling.

We thank the reviewer for this thoughtful comment and feedback.

Overall, I think this work explores well the changes in genetic-interactions over time after perturbation. The analysis is well performed and extensively benchmarked. I have the impression that the manuscript focuses almost more on the generation of the data than on the new insights.

This comment helped to further improve the quality of our manuscript. Indeed, we believe that a thorough benchmark of complex data structures and analysis based on higher-order measurements (i.e. genetic interactions) is paramount to gain certainty on the data quality for biological conclusions. Thus, in the revised manuscript we edited the manuscript to be more easily accessible and clear on its implications for biology. Specifically, we edited the Introduction and Results sections to lower the entry barrier to the manuscript.

It was also not very easy to follow but that is also due to the complexity of the experiment – non linear effects of pairs of gene knock-downs over time plus and minus drug. Even for researchers used to thinking about genetic interactions the results of these assays should be difficult to conceptualize. In part, this complexity also makes it difficult to extract novel findings on the genetic architecture of the cell.

We discussed the revised text with researchers that are not specialized on genetic interactions to make the text more accessible to other scientists despite the admittedly high complexity of the underlying data. To this end we reworked the introduction, Figure 1 and Figure 2 together with their according textual descriptions. There we specifically highlighted the conceptual questions and advances of this study (Introduction paragraph three and paragraph two “A chemo-genetic screen identifies genes sensitive to small molecule MEK inhibition”). In addition, we greatly extended the description and discussion of biologically interesting findings that highlight unexpected inter-pathway crosstalk uncovered by our experiment in the Results section (subsection Examples of context-dependent genetic interactions” paragraph two and Discussion section). Our manuscript further sets ground for future analyses and also provides all source data and code to explore and re-analyze the data.

I have no major concerns regarding the work, only a few minor points on clarification that the authors may consider:– From my reading of the manuscript the major novel findings of this work relate to the time dependencies on the differential genetic interactions. However, almost half of the Results section relate to the process of obtaining the differential interactions. While I commend the authors for the thorough characterization of the chemical-genetic and genetic-interaction data it would be possible to summarize further the initial results, in particular the initial screen for MEK inhibitor sensitive genes and to a lesser extent the establishment of the genetic interactions. This could give more space to focus on the main novel findings regarding the differential genetic interactions. The section on the differential interactions could in turn be broken down into more sub-sections trying to give more visibility to the main new findings that derive from this data.

We thank for these suggestions and edited the manuscript accordingly. Specifically, we moved parts discussing technical details to the Materials and methods. We furthermore re-structured Figure 1, 2, 4 and 5 to allow readers to more easily follow the main messages in the manuscript. A more precise definition of the different types of interactions as outlined in Figure 1 also facilitates the accessibility of the manuscript. In addition, in the Results section we now highlighted more biological insights that can be drawn from our data.

– The article gets confusing in the nomenclature of the genetic-interactions. These interactions can be: time (in)dependent, differential, stable, positive, negative, aggravating and alleviating. In particular, calling a time dependent interaction as "stable" is awkward. Perhaps using condition independent or condition insensitive instead of stable would be clearer.

We now introduced a more consistent phrasing of different context dependent genetic interactions and added precise definitions outlined in a revised Figure 1.

– The differential genetic interaction method is not described in the Materials and methods section. Are all genetic interaction pairs tested for differential interactions?

In response we shifted an expanded description of the two-factor robust linear modeling to the Materials and methods. All genes that we tested in the pair-wise interaction screen were subjected to differential interaction analysis.

Can a genetic interaction be called as significantly changing without having a strong effect size in any condition? Looking at the genetic interaction scores of replicates in Figure 2—figure supplement 5, the values range from around -5 to 5 with strong variance around -0.5 to 0.5. However, in several places in the article the interaction values and changes highlighted are often in the ranges of 0 to 0.5. Are these interaction terms calculated in the same way in Figure 2—figure supplement 5? It would appear that the variance in replicates is higher than the apparent precision of the method.

Indeed, we find an uneven distribution of variance along the absolute mean of the π-score that peaks at medium to low π-scores. Interestingly, this distribution is different for independent features and e.g. the variance for DNA eccentricity peaks at high absolute values. We are convinced that by our statistical measures (moderated t-test, alongside stringent control of the false discovery rate) we can control these effects to robustly identify biological signal also from observations in the range of low absolute scores and high variance. As a result, most interactions with low π-scores are disregarded when analyzing interactions in isolated conditions. However, as we demonstrate at multiple points throughout the manuscript, the change in π-scores due to changing conditions (time and/or treatment) can often be statistically valid, even though it’s not significant in one single condition. An example is a synthetic lethal interaction between *Rel* and pnt (Figure 8). While the interaction is weak and non-significant in either condition, the difference between conditions is however surprisingly stable across replicates and forms one of the most significant differential interaction. As we score interaction differences between conditions in contrast to their amplitude in isolated conditions we can identify significant differential interactions in many cases (Figure 3B) where single conditions were not significant. Our manuscript thus presents a strategy to robustly detect interaction differences even if the interaction strength in one isolated condition was weaker. We adopted the new “Modelling of treatment sensitive interactions” section of the Materials and methods accordingly.

– How are condition independent interactions ("stable") defined ? A cut-off on the condition sensitivity parameter ?

Condition independent interactions are defined as interactions where the treatment coefficient has no significant predictive power (FDR>0.1) while the time does influence the π-score significantly (FDR<0.1, subsection “Modelling of treatment sensitive interactions”).

– Given that the differential interactions are being called with a linear model on the time differences with a term for the drug effect, are time invariant but drug dependent genetic interactions well captured by this model ?

This kind of time-independent but treatment sensitive interactions are indeed captured by the model and often include interactions that are non-significant in either condition but their differential change was significantly measurable (Figure 1, 4, Materials and methods). This includes the example interaction between *Rel* and *pnt*. Notably, the model does not readily capture interactions that are time and treatment independent but significant in the single conditions. These need to be captured using alternative, less sensitive methods such as the one sample moderated t-test used in earlier studies (e.g. Fischer et al., 2015). To clarify this further we extended the Materials and methods section devoted to the linear modeling and also discussed its current limits (e.g. its restriction to two modelled parameters per interaction) in the discussion.

Minor Comments:- Figure 1E – annotate in the figure what are the Ras pathway components and translational regulators groups described in the Results section for clarity.

We have added the requested figure annotation.

- Figure 4F – could be move to supplementary materials and corresponding results description shortened.

We moved Figure 4F to the supplement and split Figure 4 into two figures. We thus split the text into more subsections to better guide the reader and in addition shortened technical details.

- Figure 2—figure supplement 2 – figure title is wrong

We have made the appropriate changes.